# PSG-Nav: Probabilistic Scene Graph Navigation via Multiverse Decision Making

**Rufeng Chen** [* 1]  **Yue Chang** [* 1]  **Xiaqiang Tang** [1]  **Hechang Chen** [2]  **Sihong Xie** [† 1]

## Abstract

Open-vocabulary navigation requires embodied agents to manage significant perception uncertainty stemming from semantic ambiguity and model errors. However, most existing works settle for local optimal deterministic approaches, depriving complex navigation decision-making over multiple composite possibilities that are critical for globally better solutions. In this paper, we propose Probabilistic Scene Graph Navigation (PSG-Nav), which constructs a 3D Probabilistic Scene Graph that uses full semantic categorical distributions to account for perception uncertainty. To efficiently use the local distributions to compose and reason about the optimal navigation landmarks, we propose Multiverse Decision to sample multiple most likely world settings from the joint distribution, and evaluate navigation landmarks based on the compatibility between landmarks and multiverses. To mitigate false positives due to epistemic uncertainty in open-vocabulary navigation, we introduce the Evidential Experience Calibrator, which enables online lifelong adaptation by cross-validating detections against memories of past successes and failures. Extensive experiments on widely-used benchmarks MP3D, HM3D, and HSSD demonstrate that PSG-Nav establishes new state-of-the-art results, achieving Success Rates of 66.1%, 44.8%, and 67.9%, respectively. Code is available at: https://psg-nav.github.io/

## 1. Introduction

Open vocabulary navigation in complex, unknown environments requires embodied agents to contend with the perception uncertainty due to the semantic ambiguity of the

*Equal contribution , † Corresponding Author [1]The Hong Kong University of Science and Technology (Guangzhou), Guangdong, China [2]Jilin University, Jilin, China. Correspondence to: Sihong Xie <sihongxie@hkust-gz.edu.cn>.

*Proceedings of the $43^{rd}$ International Conference on Machine Learning*, Seoul, South Korea. PMLR 306, 2026. Copyright 2026 by the author(s).

physical world and perception model errors (Gong et al., 2025; Yokoyama et al., 2024). Effectively managing this uncertainty is critical for task success. Without uncertainty awareness, the agent is forced to treat noisy perceptual outputs as absolute facts, permanently recording them in maps such as scene graphs or voxel maps. These errors then propagate through downstream planning, causing irreversible information loss and overconfident decisions that result in unreliable paths.

Current frameworks leverage Foundation Models and 3D Scene Graphs (3DSGs) to perceive and organize environmental information into structured concepts (Yin et al., 2025; Zhang et al., 2021). Advanced methods such as SG-Nav (Yin et al., 2024) and ASCENT (Gong et al., 2025) use these maps to help robots reason about object locations in complex layouts. To ensure LLM compatibility and efficiency, these methods discard full semantic distributions in favor of single fixed labels, sacrificing the probabilistic depth essential for coherent spatial reasoning. As illustrated in Figure 1 (b), this loss of information creates inaccurate maps with **illogical layouts** that lack semantic coherence. Since high-level planners rely on the map for commonsense reasoning, these spatial inconsistencies lead directly to **unreliable decision**. Ultimately, the inherent domain shift between real-world pre-training and simulated mesh renderings leads to frequent misidentifications, such as mistaking a sofa for a bed. These perceptual errors result in **false positives**, where the agent terminates the episode prematurely near an incorrect goal.

To overcome these limitations, we introduce the **3D Probabilistic Scene Graph** (3D-PSG), a hierarchical representation organized across *objects, groups,* and *rooms* that preserves the full categorical distribution for every object node (Sec. 3.2). Unlike traditional maps (Gu et al., 2024; Jatavallabhula et al., 2023; Chang et al., 2026), the 3D-PSG acts as a probabilistic buffer to rectify illogical layouts through a group-level refinement process. During the construction of group nodes, the 3D-PSG enumerates potential semantic combinations of underlying objects and leverages LLMs to prune configurations with semantic conflicts. By filtering out logically inconsistent pairings (*e.g.,* a bed appearing within a living room), this mechanism allows the agent to retain less likely but not entirely impossible labels that align with the global scene logic while suppressing highest-

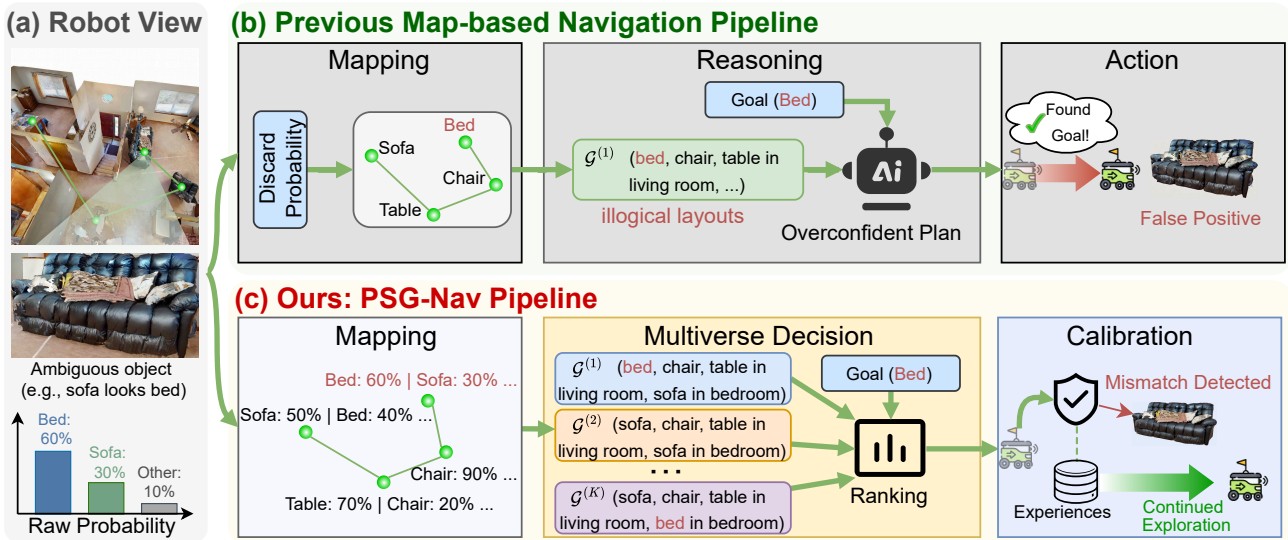

*Figure 1.* **PSG-Nav vs. Previous map-based navigation pipeline. (a)** Observations yield ambiguous semantic distributions (e.g., a sofa visually resembling a bed). **(b)** Deterministic labeling discards distributions, causing illogical layouts (e.g., a bed in a living room). This leads to overconfident reasoning and domain-shift-induced false positives, resulting in the misidentification of navigation goals. **(c)** Our approach preserves the full categorical distribution in the mapping process. The Multiverse Decision samples discrete possible worlds (world $1 \ldots K$) from the 3D-PSG's joint distribution. This enables the agent to evaluate goal utility across diverse hypotheses, effectively marginalizing perceptual noise for robust decision-making. Calibration cross-validates detections against Experiences via 3D-PSG semantic uncertainty, enabling the agent to drive Continued Exploration instead of failing prematurely.

confidence perceptual labels that are contextually incompatible with the contexts. By modeling the environment as a joint distribution factorized across the object-group-room hierarchy of the scene, the 3D-PSG ensures global semantic coherence and provides the essential evidence required for intrinsic uncertainty-aware exploration (Sec. 3.3.2).

To operationalize the 3D-PSG, we introduce **Probabilistic Scene Graph Navigation** (PSG-Nav), a unified framework that translates probabilistic representations into robust navigation actions. This framework resolves unreliable decisions through Multiverse Decision (Liu et al., 2025; Linde, 2017), which formulates navigation as a problem of decision-making under uncertainty (Sec. 3.3). By utilizing Monte Carlo sampling, this module instantiates a set of divergent possible worlds from the 3D-PSG, where each world represents a logically consistent and plausible interpretation of the environment. Within these worlds, the agent evaluates the expected utility of potential landmarks through stochastic pairwise comparisons. By aggregating results across the entire multiverse, the agent effectively marginalizes out perceptual noise and ensures decision stability. Furthermore, to mitigate false positives, we propose the **Evidential Experience Calibrator** (EEC), a Retrieval-Augmented Generation (RAG) (Lewis et al., 2020; Tanaka et al., 2025) based verification module (Sec. 3.4). The EEC maintains online memories of past successes and failures, recording visual crops alongside their room-level and relational context (Huang et al., 2025). Before goal con-

firmation, the EEC performs contrastive retrieval against these experiences to dynamically calibrate the agent's confidence. By suppressing detections that resemble known error modes while reinforcing those aligned with success patterns, the EEC ensures that the agent only terminates an episode when the goal identification is supported by hierarchical scene-graph logic and retrieved contextual consistency.

We evaluate our method on the MP3D (Chang et al., 2017), HM3D (Ramakrishnan et al., 2021), and HSSD (Khanna et al., 2024) benchmarks. Extensive experiments demonstrate that PSG-Nav establishes new state-of-the-art results. PSG-Nav achieves Success Rates of 66.1%, 44.8%, and 67.9%, respectively, while our zero-shot variant (63.5%, 43.3%, 66.1%) consistently outperforms recent baselines. We further validate PSG-Nav on a physical robot in real-world indoor environments, demonstrating its practical deployability under perceptual ambiguity.

## 2. Related Works

### 2.1. Object Goal Navigation

Object goal navigation (ObjectNav) requires the agent to locate specific categories in unseen environments. Foundational works utilized end-to-end Reinforcement (Hwang et al., 2024; Zhang et al., 2021; Mousavian et al., 2019) or Imitation Learning (Ramrakhya et al., 2022; 2023) to map observations directly to actions. However, these implicit methods (Wijmans et al., 2019; Maksymets et al., 2021)

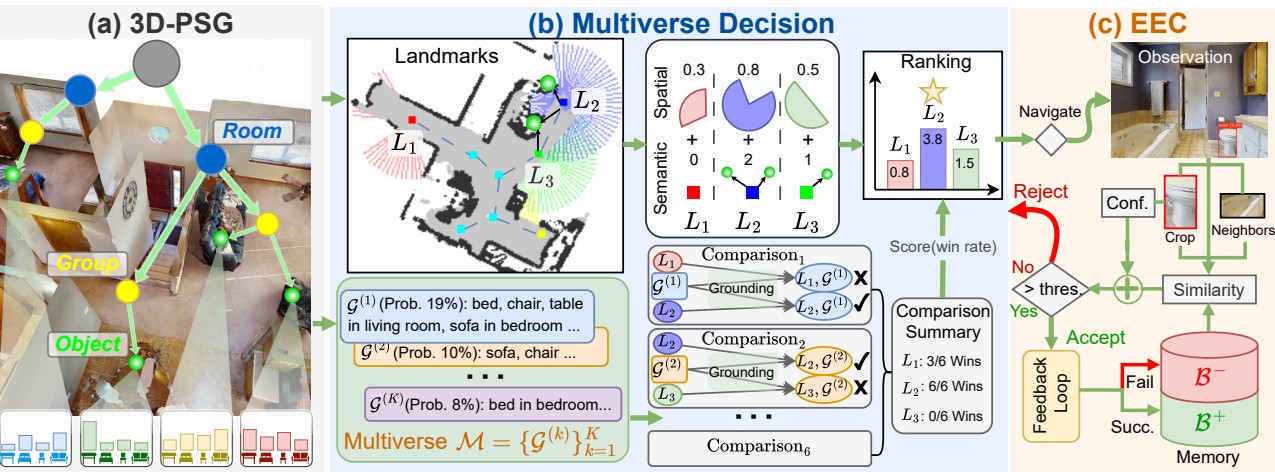

*Figure 2.* Overview of the Probabilistic Scene Graph Navigation (PSG-Nav) Framework. **(a) 3D-PSG:** We construct a probabilistic graph where objects, groups, and rooms maintain semantic distributions rather than fixed labels. **(b) Multiverse Decision:** To resolve semantic ambiguity, we sample deterministic worlds from the 3D-PSG. Each candidate landmark is grounded in these sampled realizations, transforming raw geometric coordinates into semantically-enriched prompts. A LLM then evaluates the Empirical Reasoning Utility via stochastic pairwise comparisons to select the optimal navigation goal. **(c) EEC (Evidential Experience Calibrator):** Upon detecting a candidate, the agent retrieves contexts (visual crops + structural neighbors) from success and failure memories ($\mathcal{B}^+/\mathcal{B}^-$). Detection confidence is calibrated based on similarity, and the candidate is accepted if the score is greater than a validation threshold.

often suffer from sample inefficiency and poor real-world generalization. Modular approaches (Chaplot et al., 2020; Ramakrishnan et al., 2022) address these efficiency and generalization bottlenecks by decoupling mapping, planning, and control, yet they frequently rely on closed-set detectors, limiting open-vocabulary generalization.

Recent research leverages Foundation Models for zero-shot alignment (Majumdar et al., 2022) and commonsense-driven exploration (Zhou et al., 2023; Yu et al., 2023). To provide persistent environmental abstraction, modular frameworks like SG-Nav (Yin et al., 2024) and CogNav (Cao et al., 2025) prompt over 3D Scene Graphs (3DSGs) or maintain dynamic cognitive states. Building on these foundations, new SOTA methods address increasingly complex geometric constraints: ASCENT (Gong et al., 2025) enables multi-floor navigation through stair-aware, coarse-to-fine exploration, ApexNav (Zhang et al., 2025) enhances efficiency via goal-centric semantic fusion, and BeliefMapNav (Zhou et al., 2026) utilizes 3D voxel-based belief mapping for refined spatial priors. However, despite these structural and geometric advancements, existing frameworks predominantly rely on deterministic representations.

### 2.2. Uncertainty in Embodied Decision Making

Uncertainty modeling is essential for both robust perception and reliable planning. Traditional mapping methods utilize factor graphs or Bayesian kernels to model metric-semantic and pose uncertainty (Bowman et al., 2017; Gan et al., 2020). Recent advances employ Conformal Prediction

to provide statistically grounded scene graph estimates with rigorous confidence intervals (Nag et al., 2025). Building on these representations, navigation frameworks ground LLM confidence in physical affordances (Brohan et al., 2023) or utilize calibration sets for safety guarantees (Ren et al., 2023; Yin et al., 2026). More advanced methods incorporate formal decision theory, optimizing for expected utility or entropy reduction to actively resolve environmental ambiguity (Hu et al., 2024; Liu et al., 2025). However, unifying hierarchical probabilistic structures with open-vocabulary reasoning remains an open challenge. We address this by modeling a 3D-PSG as a joint distribution and resolving it via Multiverse Decision-making.

## 3. Method

We propose **Probabilistic Scene Graph Navigation**, a framework designed to robustly navigate under semantic ambiguity and perception uncertainty in both simulation and real-world settings. As illustrated in Figure 2, our system consists of three core components: (A) a 3D Probabilistic Scene Graph (3D-PSG) that explicitly models semantic distributions (Sec. 3.2); (B) a Multiverse Decision framework that samples worlds for reliable planning (Sec. 3.3); and (C) an Evidential Experience Calibrator that calibrates decision confidence using historical context (Sec. 3.4).

### 3.1. Problem Formulation

We address the Continual Open-Vocabulary Object Navigation task, where an agent must locate objects in unseen

environments and adapt to perception errors over time. The goal category $c$ is provided as free-form text (e.g., "blue sofa"). At each timestep $t$, the agent receives a posed RGB-D observation $O_t = \{I_t^{rgb}, I_t^{depth}, p_t\}$, where $I_t^{rgb}$ and $I_t^{depth}$ are the color and depth images, and $p_t = (x_t, y_t, \theta_t)$ denotes the agent's current position and orientation from odometry. Based on the observation history, the agent selects and executes an action $a_t$ from a discrete action space $\mathcal{A} = \{\text{MOVE\_FORWARD}, \text{TURN\_LEFT}, \text{TURN\_RIGHT}, \text{LOOK\_UP}, \text{LOOK\_DOWN}, \text{STOP}\}$. An episode is successful if the agent executes the STOP action within a threshold distance $d_s$ of the goal object while it remains visible, all within a maximum time budget of $T_{max}$ steps.

## 3.2. 3D Probabilistic Scene Graph

To represent the complex spatial and semantic environment, we define the 3D Probabilistic Scene Graph (3D-PSG) at time $t$ as a hierarchical multi-layered graph $\mathcal{G}_t = (\mathcal{V}, \mathcal{E})$. The vertex set $\mathcal{V} = \{O, G, R\}$ is partitioned into three semantic tiers: Object nodes ($O = \{o_1, \ldots, o_{N_O}\}$), Group nodes ($G = \{g_1, \ldots, g_{N_G}\}$), and Room nodes ($R = \{r_1, \ldots, r_{N_R}\}$). Unlike traditional deterministic scene graphs (*e.g.*, SG-Nav (Yin et al., 2024)) that assign a single hard label to each entity, every object node $v \in \mathcal{V}$ in our 3D-PSG maintains a full categorical distribution $\mathbf{p}_v \in \Delta^{C-1}$ over $C$ semantic categories. The edge set $\mathcal{E}$ represents hierarchical containment and spatial proximity, where an edge $e_{ij} \in \mathcal{E}$ exists if node $v_i$ is spatially encapsulated by or functionally related to node $v_j$ in a higher tier. This probabilistic formulation allows the agent to preserve alternative semantic hypotheses across the object-group-room hierarchy, effectively preventing the irreversible information loss caused by deterministic labels during initial perception.

### 3.2.1. PROBABILISTIC OBJECT NODES

Each object node $o_i$ in our 3D-PSG is associated with a dynamic belief over the semantic category set $\mathcal{C}$. To capture and maintain perception uncertainty throughout the navigation episode, we maintain a dedicated count vector $\mathbf{n}_{i,t} \in \mathbb{R}^{|\mathcal{C}|}$ for each object $o_i$, where the $k$-th element $n_{i,t}^{(k)}$ represents the accumulated evidence for category $c_k \in \mathcal{C}$ up to time $t$. At each timestep $t$ where object $o_i$ is observed, the perception model provides a predicted label vector $\mathbf{v}_{i,t}$ (*e.g.*, a one-hot vector of the highest-confidence class). The belief is then updated and normalized as:

$$\mathbf{n}_{i,t} = \mathbf{n}_{i,t-1} + \mathbf{v}_{i,t}; \quad P_t(o_i = c_k) = \frac{n_{i,t}^{(k)}}{\sum_{j=1}^{|\mathcal{C}|} n_{i,t}^{(j)}}. \quad (1)$$

While Bayesian updates with Dirichlet priors are theoretically attractive, open-vocabulary detectors often exhibit poor probability calibration, rendering confidence accumu-

lation unstable. We therefore adopt this robust "vote" accumulation strategy (Eq. 1).

This approach effectively filters transient noise by treating detections as discrete votes, while simultaneously preserving the full categorical distribution to serve as a robust foundation for the subsequent multiverse decision process.

### 3.2.2. HIERARCHICAL PROBABILISTIC SCENE GRAPH

Defining a joint distribution over all objects leads to a combinatorial explosion (*e.g.*, even the most likely global configuration often holds $< 10\%$ probability), making reasoning intractable. To mitigate this, we factorize uncertainty via a hierarchical structure by clustering spatially proximal and semantically correlated objects into Group nodes. For each group node $g_j$ containing $N_j$ objects, the probability of a specific semantic configuration $s$ is computed as the product of the independent beliefs of its constituent objects:

$$P(g_j = s) = \prod_{i=1}^{N_j} P(o_{j,i} = c_{j,i}^s), \quad (2)$$

where $c_{j,i}^s$ is the semantic category of object $o_{j,i}$ in configuration $s$. For instance, if $o_{j,1}$ is (70% table, 30% desk) and $o_{j,2}$ is (80% chair, 20% stool), $g_j$ stores the joint probabilities for all outcomes (*e.g.*, $P(\text{table, chair}) = 0.56$).

Building upon this, room-level hypotheses are formed by aggregating the distributions of their child groups. This hierarchical factorization (Object $\to$ Group $\to$ Room) enables a structured representation of environmental uncertainty without the overhead of global joint modeling. This provides the probabilistic foundation for the downstream Multiverse Decision process described in Sec. 3.3.

## 3.3. Multiverse Decision

To enable effective reasoning over the 3D-PSG, we introduce a Multiverse Decision Framework that approximates its hierarchical joint distributions by sampling a set of discrete worlds $\mathcal{M} = \{\mathcal{G}^{(1)}, ..., \mathcal{G}^{(K)}\}$. Each world $\mathcal{G}^{(k)}$ represents a consistent, deterministic instantiation of the environment's semantic state. This formulation transforms intractable probabilistic planning into a series of concrete reasoning tasks, allowing the agent to explicitly ground uncertainty into active exploration.

### 3.3.1. HIERARCHICAL MULTIVERSE CONSTRUCTION

To realize the Multiverse Decision Framework, we transform the 3D-PSG into a set of discrete worlds $\mathcal{M}$ through a hierarchical bottom-up process that integrates probabilistic enumeration with LLM-guided logical pruning.

At the group-level, we first generate semantic configurations $s$ for each group $g_j$ by enumerating constituent object-level

beliefs and retaining the top-$K_g$ most probable candidates to form the initial set $\mathcal{S}_{g_j}$. To ensure local semantic coherence, we leverage an LLM as a logical filter $f_{\text{LLM}}(\cdot) \in \{0, 1\}$ to yield a refined set:

$$\hat{\mathcal{S}}_{g_j} = s \in \mathcal{S}_{g_j} \mid f_{\text{LLM}}(s) = 1, \tag{3}$$

where $f_{\text{LLM}}(s) = 0$ if the configuration exhibits internal logical conflicts (*e.g.*, toilet appearing within living room).

Similarly, room-level hypotheses are formed by combining the refined group configurations within each room $r_k$. From the resulting combinations, we retain the top-$K_r$ candidates and again apply the LLM filter to discard those that violate architectural common sense or spatial logic.

This dual-filtering strategy suppresses combinatorial noise and ensures that the agent reasons only over structurally sound environmental hypotheses, preventing the propagation of perceptual errors into global navigation decisions.

### 3.3.2. INTRINSIC UNCERTAINTY-AWARE EXPLORATION

To bridge the gap between static mapping and active navigation, we discretize the environment into a candidate landmark set $\mathcal{L}_t = \{l_{i,t}\}$ derived from the Generalized Voronoi Graph (GVG) and geometric frontiers. (*See Appendix A.3.1 for implementation details.*) Before performing goal-specific reasoning, the agent evaluates the epistemic value of each landmark by simulating the information it would acquire if it were to occupy that location. This projected utility is defined as a weighted sum of spatial and semantic components: $U_{\text{gain}}(l_{i,t}) = \alpha \cdot I_{\text{spa}}(l_{i,t}) + I_{\text{sem}}(l_{i,t})$

**Spatial Information Discovery.** The geometric utility $I_{\text{spa}}(l_i)$ is motivated by the objective of maximizing the perceptual information yield of a navigational action. The agent simulates the incremental coverage of unknown space $\mathcal{U}$ that would be observable upon reaching landmark $l_{i,t}$:

$$I_{\text{spa}}(l_{i,t}) = \frac{|\mathcal{U}(l_{i,t})|}{\pi r_{\max}^2} \tag{4}$$

where $r_{\max}$ is the agent's maximum sensing range and $|\mathcal{U}(l_{i,t})|$ denotes the set of unknown regions observable from $l_{i,t}$. This metric prioritizes landmarks that resolve the most significant geometric blind spots in the occupancy map. (*See Appendix A.3.2 for implementation details.*)

**Semantic Disambiguation Potential.** The semantic component $I_{\text{sem}}(l_{i,t})$ is grounded in the physical intuition that proximity yields clarity. Intuitively, since perception noise and categorical ambiguity decrease as the agent approaches an object, the agent evaluates the expected reduction in semantic uncertainty at $l_{i,t}$. By aggregating the Shannon entropy of proximal objects $\mathcal{O}_p(l_{i,t})$, where we denote $\mathcal{O}_p(l_{i,t})$ as $\mathcal{O}_p$ for brevity. The agent identifies highly uncertain regions where close-range observations can most effectively resolve the semantic state of the 3D-PSG:

$$I_{\text{sem}}(l_{i,t}) = - \sum_{o_i \in \mathcal{O}_p} \sum_{c \in \mathcal{C}} P_t(o_i = c) \log P_t(o_i = c) \tag{5}$$

### 3.3.3. PROBABILISTIC DECISION

Given the candidate landmark set $\mathcal{L}_t$ in Sec. 3.3.2 and their respective information gains, the decision module selects an optimal local goal that maximizes the probability of successfully locating the goal object. To maintain computational efficiency and focus the LLM's reasoning on the most promising regions, we implement a low-cost epistemic pruning step.

We define a filtering threshold $\tau$ to retain only high-potential landmarks $l_{i,t} \in \mathcal{L}_t$ that offer significant exploratory value:

$$\mathcal{L}'_t = l_{i,t} \in \mathcal{L}_t \mid U_{\text{gain}}(l_{i,t}) \geq \tau \tag{6}$$

This filtering mechanism effectively eliminates redundant candidates, such as previously visited areas or semantically unambiguous empty spaces. By concentrating the subsequent LLM-based reasoning on the refined set $\mathcal{L}'_t$, we ensure that the computational budget is exclusively allocated to frontiers that promise substantial spatial or semantic refinement of the 3D-PSG.

**Stochastic Pairwise Comparison.** To navigate robustly under semantic uncertainty, the agent must evaluate landmarks not as isolated points, but as strategic nodes within a set of possible environmental configurations. We treat the final goal selection as an Expected Utility Maximization problem, where the utility of a landmark is its potential to lead the agent to the goal $g$. Rather than reasoning over a single, noisy scene representation, we perform $M$ rounds of evaluation. In each round $m \in \{1, \ldots, M\}$, a deterministic scene graph $\mathcal{G}^{(m)}$ is sampled from the multiverse $\mathcal{M}$. For each filtered landmark $l_{i,t} \in \mathcal{L}'_t$, we generate a context-aware descriptor $D(l_{i,t} \mid \mathcal{G}^{(m)})$ by retrieving its proximal objects and room-level semantics within the specific world $\mathcal{G}^{(m)}$. This ensures the LLM reasons over a concrete and structurally consistent environment in every instance. To mitigate the position bias and sensitivity inherent in LLM-based ranking, we employ a stochastic pairwise comparison strategy. For each world $\mathcal{G}^{(m)}$, the LLM acts as a preference oracle $f_{\text{LLM}}(\cdot)$ that compares two landmarks $l_i$ and $l_j$:

$$\mathbb{I}(l_i \succ l_j \mid \mathcal{G}^{(m)}) = f_{\text{LLM}}(D(l_i \mid \mathcal{G}^{(m)}), D(l_j \mid \mathcal{G}^{(m)}), g) \tag{7}$$

where $\mathbb{I} = 1$ if $l_i$ is judged more conducive to finding goal $g$ than $l_j$, and 0 otherwise.

**Pairwise Preference Oracle.** The final preference score $S(l_{i,t})$ for each landmark is derived by marginalizing these

pairwise wins across the entire multiverse:

$$S(l_{i,t}) = \frac{1}{M \cdot (|\mathcal{L}'_t| - 1)} \sum_{m=1}^{M} \sum_{j \neq i} \mathbb{I}(l_{i,t} \succ l_{j,t} \mid \mathcal{G}^{(m)})$$

(8)

The optimal local goal $l^*$ is then selected by combining this semantic preference with the intrinsic information gain: $l^* = \arg\max(S(l_{i,t}) + \beta U_{\text{gain}}(l_{i,t}))$. This formulation ensures the agent's trajectory is both goal-oriented (guided by the multiverse consensus) and uncertainty-aware exploration (driven by the need to resolve uncertainty).

### 3.4. Evidential Experience Calibrator

Standard object detection models are often prone to persistent false positives, where objects with similar visual features are consistently misidentified. To prevent the agent from terminating its search at an incorrect location, we introduce the Evidential Experience Calibrator (EEC), a RAG-based verifier to calibrate stopping decisions by cross-referencing visual detections with contextual evidence.

**Probabilistic Context Memory.** Unlike traditional RAG methods that store isolated image crops, we construct a persistent memory of probabilistic context descriptors. We maintain two memories: a Positive Bank ($\mathcal{B}^+$), containing descriptors of successfully identified goals, and a Negative Bank ($\mathcal{B}^-$), storing descriptors of historical false positives. Each entry $m$ stores a composite embedding $m = (\mathbf{v}_{\text{vis}}^m, \mathbf{v}_{\text{struct}}^m)$ in memories. The first component, $\mathbf{v}_{\text{vis}}^m$ encodes the visual appearance of the object. Crucially, the second component, $\mathbf{v}_{\text{struct}}^m$ encodes the scene context as a tuple of two distributions: $\mathbf{v}_{\text{struct}}^m = (p_{\text{R}}^m, p_{\text{G}}^m)$. Specifically, $p_{\text{R}}^m \in \mathbb{R}^{|\mathcal{C}_{\text{room}}|}$ represents the object's room semantic distribution, and $\mathbf{p}_{\text{G}}^m$ aggregates the semantic distributions of all other objects within the same group. This dual-distribution encoding allows the system to capture both global architectural context and local co-occurrence patterns.

**Retrieval-Based Confidence Calibration.** To ensure robust stopping criteria, we implement a bi-directional probabilistic calibration mechanism modulated by semantic uncertainty. When the agent identifies a candidate object $o_c$ with an initial detection score $S_{\text{det}}$, we characterize it by its descriptor $(\mathbf{v}_{\text{vis}}, p_{\text{G}}, p_{\text{R}})$ and query the memory banks using a hybrid similarity metric. Specifically, visual features $\mathbf{v}_{\text{vis}}$ and aggregated group contexts $p_{\text{G}}$ are treated as semantic embeddings compared via Cosine Similarity, while the room context $p_{\text{R}}$ is treated as a discrete probability distribution compared via Jensen-Shannon Divergence (JSD). The composite similarity is computed as:

$$\begin{aligned} \text{sim}(o_c, m) = \cos(\mathbf{v}_{\text{vis}}, \mathbf{v}_{\text{vis}}^m) + w_1 \cdot \cos(p_{\text{G}}, p_{\text{G}}^m) \\ + w_2 \cdot (1 - \text{JSD}(p_{\text{R}}, p_{\text{R}}^m)), \end{aligned}$$

(9)

where $w_1, w_2$ are balancing weights. Based on this metric,

we retrieve the nearest matches from both banks:

$$S_{\text{pos}} = \max_{m \in \mathcal{B}^+} \text{sim}(o_c, m); \quad S_{\text{neg}} = \max_{m \in \mathcal{B}^-} \text{sim}(o_c, m). \quad (10)$$

We then compute a calibration margin $\Delta S = S_{\text{pos}} - \gamma \cdot S_{\text{neg}}$, where $\gamma$ weighs the penalty of retrieving a negative match.

Rather than relying on complex modulation, we directly refine the raw confidence by adding this retrieval-based evidence: $S_{\text{final}} = S_{\text{det}} + \Delta S$. Finally, the agent confirms the detection and terminates the episode only if the calibrated score exceeds a validation threshold: $S_{\text{final}} > \delta$.

This linear formulation effectively boosts confidence when the candidate aligns with past successes ($S_{\text{pos}} > \gamma S_{\text{neg}}$) and suppresses it when the candidate resembles known failure modes, ensuring that only high-quality detections trigger a stop action.

**Diversity-Driven Update.** The memory banks $\mathcal{B}^+$ and $\mathcal{B}^-$ are updated dynamically at the conclusion of each episode based on ground-truth feedback. To ensure computational tractability and bound storage growth, we enforce a capacity limit $N_{\text{max}}$ for each bank. When the bank reaches capacity, we implement a redundancy-aware pruning strategy: the entry with the highest average similarity to all other samples in the bank is removed. Mathematically, we discard entry $m^*$ such that:

$$m^* = \arg\max_{m \in \mathcal{B}} \frac{1}{|\mathcal{B}| - 1} \sum_{m' \in \mathcal{B}, m' \neq m} \text{sim}(m, m'). \quad (11)$$

By pruning generic data to preserve salient outliers, this strategy ensures a diverse, compact memory of informative samples essential for robust decision calibration.

## 4. Experiment

We evaluate **PSG-Nav** to investigate three central questions: (1) Does explicitly modeling semantic uncertainty improve navigation success in open-world scenarios? (2) How does each module contribute to performance? (3) Can PSG-Nav be reliably deployed on a physical robot in real-world indoor environments under perceptual ambiguity?

### 4.1. Experimental Setup

**Datasets.** We conduct evaluations on the ObjectGoal Navigation (ObjectNav) task using the Habitat simulator. We utilize standard validation splits from MP3D (Chang et al., 2017), a large-scale indoor 3D scene dataset commonly used in Habitat-based ObjectNav evaluations. HM3D (Ramakrishnan et al., 2021), the official dataset of the Habitat 2022 ObjectNav Challenge, includes 2,000 validation episodes across 20 environments and six object categories. HSSD (Khanna et al., 2024), a synthetic dataset with scenes based

*Table 1.* Comparison with SOTA methods on HM3D, MP3D, and HSSD benchmarks. We report results for both our Static (strict zero-shot) and Adaptive (online update) variants. **PSG-Nav (Adaptive)** achieves state-of-the-art Success Rates across all benchmarks by leveraging lifelong learning.

| Method | Unsupervised | Zero-shot | HM3D | | MP3D | | HSSD | |
|---|---|---|---|---|---|---|---|---|
| | | | SR (%) ↑ | SPL (%) ↑ | SR (%) ↑ | SPL (%) ↑ | SR (%) ↑ | SPL (%) ↑ |
| Habitat-Web (Ramrakhya et al., 2022) | × | × | 41.5 | 16.0 | 31.6 | 8.5 | - | - |
| OVRL (Yadav et al., 2023) | × | × | - | - | 28.6 | 7.4 | - | - |
| ProcTHOR (Deitke et al., 2022) | × | × | 54.4 | 31.8 | - | - | - | - |
| SGM (Zhang et al., 2024) | × | × | 60.2 | 30.8 | 37.7 | 14.7 | - | - |
| ZSON (Majumdar et al., 2022) | × | √ | 25.5 | 12.6 | 15.3 | 4.8 | - | - |
| PSL (Sun et al., 2024) | × | √ | 42.4 | 19.2 | 18.9 | 6.4 | - | - |
| PixNav (Cai et al., 2024) | × | √ | 37.9 | 20.5 | - | - | - | - |
| ImagineNav (Zhao et al., 2025) | × | √ | 53.0 | 23.8 | - | - | 51.0 | 24.9 |
| VLFM (Yokoyama et al., 2024) | √ | √ | 52.5 | 30.4 | 36.4 | 17.5 | - | - |
| ESC (Zhou et al., 2023) | √ | √ | 39.2 | 22.3 | 28.7 | 14.2 | 38.1 | 22.2 |
| Cows (Gadre et al., 2023) | √ | √ | - | - | 9.2 | 4.9 | | |
| L3MVN (Yu et al., 2023) | √ | √ | 50.4 | 23.1 | 34.9 | 14.5 | 41.2 | 22.5 |
| VoroNav (Wu et al., 2024) | √ | √ | 42.0 | 26.0 | - | - | 41.0 | 23.2 |
| GAMap (Yuan et al., 2024) | √ | √ | 53.1 | 26.0 | - | - | - | - |
| OpenFMNav (Kuang et al., 2024) | √ | √ | 52.5 | 24.1 | 37.2 | 15.7 | - | - |
| InstructNav (Long et al., 2024) | √ | √ | 58.0 | 20.9 | - | - | - | - |
| SG-Nav (Yin et al., 2024) | √ | √ | 54.0 | 24.9 | 40.2 | 16.0 | - | - |
| Unigoal (Yin et al., 2025) | √ | √ | 54.5 | 25.1 | 41.0 | 16.4 | - | - |
| BeliefMapNav (Zhou et al., 2026) | √ | √ | 61.4 | 30.6 | 37.3 | 17.6 | 65.2 | 32.1 |
| ApexNav (Zhang et al., 2025) | √ | √ | 59.6 | 33.0 | 39.2 | 17.8 | - | - |
| ASCENT (Gong et al., 2025) | √ | √ | 65.4 | **33.5** | 44.5 | 15.5 | - | - |
| **PSG-Nav (w/o EEC)** | √ | √ | 63.5 | 31.2 | 43.3 | 17.6 | 66.1 | 32.2 |
| **PSG-Nav** | √ | × | **66.1** | 32.1 | **44.8** | **17.9** | **67.9** | **33.4** |

on real house layouts, contains 40 validation scenes, 1,248 navigation episodes, and six object categories, and is used to test open-vocabulary generalization in indoor environments.

**Evaluation Metrics.** We report Success Rate (SR), defined as reaching within 1.0m of any oracle-visible goal instance, and Success weighted by Path Length (SPL) to assess both navigation effectiveness and path efficiency.

**Implementation Details.** We limit navigation to 500 steps and maintain a $800 \times 800$ occupancy map. We employ GLIP (Li* et al., 2022) for open-vocabulary detection, Grounded-SAM (Ren et al., 2024) for Scene graph construction, and Qwen2.5-7B-Instruct (Bai et al., 2023) as the reasoning engine. Crucially, we employ a lifelong evaluation protocol where the agent utilizes termination signals to update persistent semantic memories ($\mathcal{B}^+, \mathcal{B}^-$) across validation episodes. The number of rays $N_r = 72$ in Sec. 3.3.2. The multiverse sampling count is set to $K = 3$, $\tau = 0.1$ and the $\alpha = 1, \beta = 0.5$ in the Sec. 3.3. We set the capacity limit of Evidential Experience Calibrator $N_{\max} = 10$, and $\gamma = 2$, $\delta = 0.61$, in the Sec. 3.4.

## 4.2. Main Results

As shown in Table 1, PSG-Nav establishes new state-of-the-art results in both zero-shot and lifelong object navigation settings. On the large-scale HM3D dataset, it achieves

*Table 2.* Ablation study on 3D-PSG and hierarchical structure.

| Method | HM3D | | MP3D | | HSSD | |
|---|---|---|---|---|---|---|
| | SR↑ | SPL↑ | SR↑ | SPL↑ | SR↑ | SPL↑ |
| w/o 3D-PSG | 58.4 | 25.3 | 42.5 | 16.2 | 58.5 | 30.7 |
| w/o Group | 58.8 | 25.3 | 43.1 | 16.4 | 59.9 | 30.9 |
| w/o Room | 59.7 | 25.6 | 43.6 | 16.6 | 61.7 | 31.3 |
| **PSG-Nav** | **66.1** | **32.1** | **44.8** | **17.9** | **67.9** | **33.4** |

66.1% SR, surpassing the deterministic baseline SG-Nav by a massive margin (+12.1%). Crucially, even in the strict zero-shot setting, the variant (w/o EEC) achieves 63.5% SR. This outperforms competitive methods like BeliefMapNav (61.4%) and ApexNav (59.6%), falling slightly short only of ASCENT (65.4%), which explicitly benefits from floor-aware exploration capabilities. On the challenging MP3D and HSSD datasets, PSG-Nav maintains this dominance (44.8% and 67.9% SR, respectively), indicating that our uncertainty-aware planning prevents the agent from getting stuck in local optima compared to other baselines.

## 4.3. Ablation Study

To validate the contribution of each component, we conduct comprehensive ablation studies on all three benchmarks. (1) the effect of the 3D probabilistic scene graph and its hierarchical structure; (2) the contribution of the Multiverse

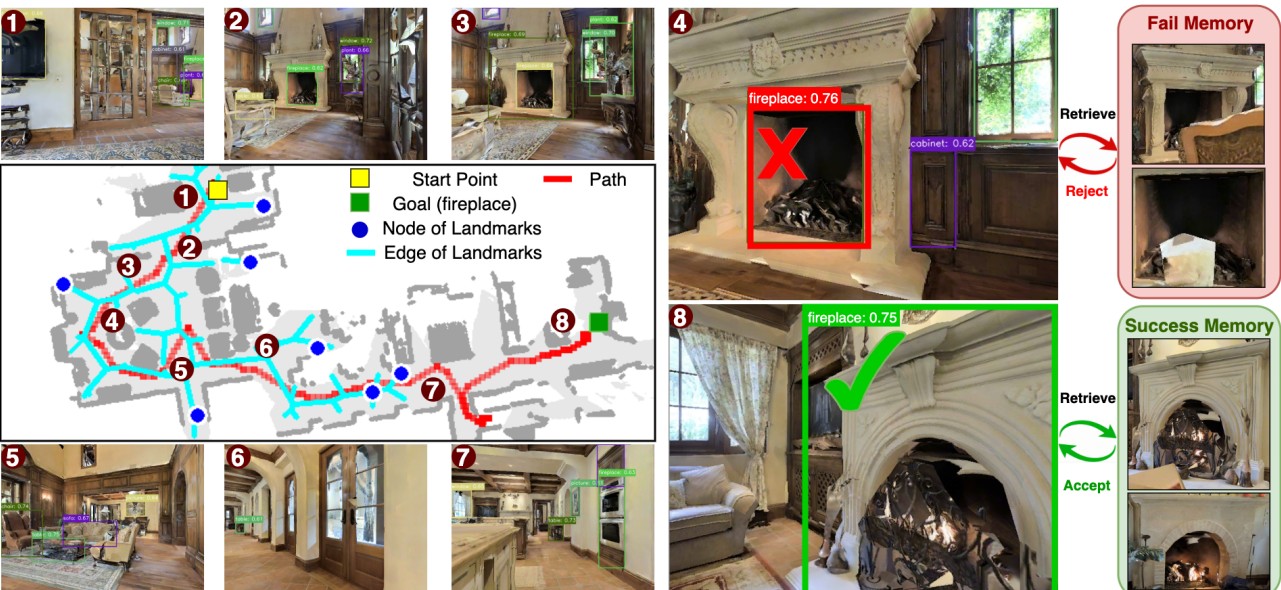

*Figure 3.* Qualitative visualization of the PSG-Nav navigation process and the Evidential Experience Calibrator (EEC). The agent is tasked with locating a "fireplace." It first encounters a visually ambiguous false positive. By cross-referencing the candidate's spatial and semantic context with the Fail Memory ($\mathcal{B}^-$), the EEC correctly rejects the detection, effectively preventing premature termination. Driven by continued exploration, the agent locates the true goal, which is successfully validated against the Success Memory ($\mathcal{B}^+$).

*Table 3.* Ablation study on the Multiverse Decision module. We investigate the impact of Information Gain components, the reasoning strategy, and the multiverse sampling density ($K$).

| Method | HM3D | | MP3D | | HSSD | |
|---|---|---|---|---|---|---|
| | SR↑ | SPL↑ | SR↑ | SPL↑ | SR↑ | SPL↑ |
| *Information Gain* | | | | | | |
| w/o Spa. & Sem. | 62.1 | 29.2 | 41.2 | 16.7 | 63.1 | 32.8 |
| w/o Spa. | 64.1 | 30.5 | 43.6 | 16.9 | 66.2 | 32.9 |
| w/o Sem. | 65.5 | 31.6 | 44.0 | 17.5 | 66.9 | 33.2 |
| *Reasoning Mechanism* | | | | | | |
| w/o Comparison | 63.7 | 30.1 | 42.5 | 17.0 | 63.5 | 32.3 |
| Ranking | 59.9 | 25.4 | 42.7 | 16.4 | 62.7 | 29.8 |
| *Multiverse Sampling Density* | | | | | | |
| $K = 1$ | 63.3 | 31.1 | 43.5 | 17.1 | 65.7 | 32.3 |
| $K = 2$ | 65.1 | 31.8 | 44.5 | 17.6 | 67.2 | 32.9 |
| **PSG-Nav** ($K = 3$) | **66.1** | **32.1** | **44.8** | **17.9** | **67.9** | **33.4** |

Decision module; and (3) the effectiveness of the Evidential Experience Calibrator under different memory strategies.

**Impact of Probabilistic Hierarchy.** Table 2 confirms the necessity of both uncertainty and structure. Reverting to deterministic labels (w/o 3D-PSG) causes a sharp drop (e.g., -9.4% SR on HSSD). Notably, removing Group nodes (w/o Group) performs nearly as poorly as the deterministic baseline (58.8% vs. 58.4% SR on HM3D). This confirms that without hierarchical factorization, the combinatorial explosion of flat object states makes the joint probability of any global configuration negligible, rendering the sampling of valid worlds intractable.

**Ablation on Multiverse Decision.** The effectiveness of the Multiverse Decision module is analyzed in Table 3. We observe that the synergy between spatial and semantic information gain is fundamental; removing both components (w/o Spa. & Sem.) results in a 4.0% drop in SR on HM3D, highlighting the necessity of combining geometric exploration with active semantic ambiguity resolution. Furthermore, our stochastic pairwise comparison strategy proves far more robust than direct absolute scoring (Ranking), which suffers a significant 6.2% SR decrease on HM3D. This underscores the advantage of leveraging the LLM's strength in relative preference over uncalibrated absolute utility estimation. Finally, performance scales with the multiverse sampling density $K$, where increasing $K$ from 1 to 3 yields a 2.8% SR gain on HM3D. This trend validates that marginalizing over environmental uncertainty through multiple sampled "worlds" leads to a more robust consensus.

**Ablation on Evidential Experience Calibrator.** To evaluate the efficacy of the EEC in mitigating perception errors, we compare two strategic variants: (1) removing the module entirely (w/o EEC); (2) utilizing a pre-filled **Static Memory** (10 samples per category, initialized via an offline warm-up episode). As indicated in Table 4, the absence of the EEC leads to a significant performance degradation on HM3D (63.5% vs 66.1%), highlighting the critical need for calibration. Notably, employing a Static Memory alone yields a substantial improvement, confirming that retrieving from the semantic context helps filter false positives.

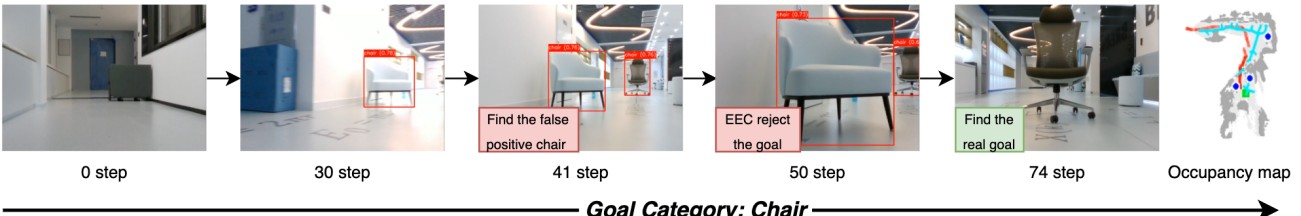

*Figure 4.* Real-world deployment of PSG-Nav on a physical robot. The robot searches for a chair in an indoor environment. A visually similar sofa is first misidentified as the target, but the Evidential Experience Calibrator rejects this false positive based on past experience. The robot then continues exploration and successfully reaches the true chair.

*Table 4.* Ablation study on the EEC strategies. We compare the default online dynamic updating against removing the module and using a static offline memory bank.

| Method | HM3D | | MP3D | | HSSD | |
|---|---|---|---|---|---|---|
| | SR↑ | SPL↑ | SR↑ | SPL↑ | SR↑ | SPL↑ |
| w/o EEC | 63.5 | 31.2 | 43.3 | 17.6 | 66.1 | 32.2 |
| **Static Memory** | 66.0 | 32.0 | 44.5 | 17.7 | 67.7 | 33.7 |
| **PSG-Nav** | **66.1** | **32.1** | **44.8** | **17.9** | **67.9** | **33.4** |

This error-correction capability is qualitatively demonstrated in Figure 3, the agent rejects the first fireplace candidate because its structural context matches failure cases in the Fail Memory ($\mathcal{B}^-$). It then continues exploration to correctly accept a goal that aligns with the Success Memory ($\mathcal{B}^+$). These results confirm that our framework successfully leverages the structural and probabilistic information of the 3D-PSG to enhance decision reliability.

### 4.4. Real-World Experiments

To further validate the deployability of PSG-Nav beyond simulation, we conduct qualitative experiments on a physical robot in real indoor environments. *A detailed description of the robot settings is in the Appendix A.1.*

We visualize a representative real-world deployment of PSG-Nav in Figure 4. The robot is tasked with finding a chair in an indoor environment. During navigation, the perception module first detects a visually similar sofa as a chair, producing a false positive candidate. Instead of prematurely stopping, the Evidential Experience Calibrator (EEC) rejects this candidate by leveraging past contextual experience, allowing the agent to continue exploration. The robot eventually reaches the true chair and successfully confirms the goal. This example demonstrates that PSG-Nav can mitigate real-world perception ambiguity and improve the reliability of stopping decisions beyond simulation.

### 5. Conclusion

We introduce PSG-Nav to resolve the deterministic collapse in open-vocabulary navigation by modeling environments as 3D Probabilistic Scene Graphs. To render probabilistic planning tractable, we propose the Multiverse Decision that leverages pairwise comparisons, complemented by the Evidential Experience Calibrator for confidence calibration.

**Limitations and Future Work.** A primary limitation is our reliance on 2D occupancy maps for exploration, which restricts PSG-Nav from explicitly handling vertical navigation, such as stairs, elevators, and multi-floor topologies. In addition, although landmark pruning and a small sampling budget keep multiverse reasoning tractable, repeated LLM-based pairwise comparisons may still introduce latency in large-scale environments. Future work will extend PSG-Nav toward volumetric planning and hierarchical floor-level scene graphs.

### Acknowledgements

Sihong Xie was supported by the National Key R&D Program of China (Grant No.2023YFF0725001), the Department of Science and Technology of Guangdong Province (2023CX10X079), the Guangzhou-HKUST(GZ) Joint Funding Program (Grant No.2023A03J0008), and the Education Bureau Guangzhou Municipality. Hechang Chen was supported by the National Key R&D Program of China (No. 2023YFF0905400, No. 2021ZD0112500); the National Natural Science Foundation of China (No. 62476110, No. U2341229); the National Key R&D Program of China (No. 2023YFF0905400, No. 2021ZD0112500); the Key R&D Project of Jilin Province (No. 20240304200SF); NSFC Grant (No. 62576364).

### Impact Statement

This paper presents work whose goal is to advance the field of Machine Learning by improving the robustness of autonomous navigation under perception uncertainty. Our proposed framework, PSG-Nav, enables more reliable decision-making in complex environments by preserving full semantic distributions and utilizing historical experience for calibration. We believe the advancements in uncertainty-aware navigation presented here contribute to the development of more trustworthy and capable autonomous agents.

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

# A. Appendix

## A.1. Real-World Experiments

To further validate the deployability of PSG-Nav beyond simulation, we conduct qualitative experiments on a physical robot in real indoor environments. Our robotic platform is built upon an Agilex SCOUT MINI chassis. The sensor suite comprises a RealSense D435 RGB-D camera for visual perception, a CH110 IMU, and dual Livox MID-360 LiDARs configured to provide omnidirectional geometric coverage and blind-spot reduction. All data processing and sensor synchronization are performed onboard via an NVIDIA Jetson AGX Xavier computer.

The rigid extrinsic transformations among the RGB-D camera, IMU, and dual LiDARs are pre-calibrated by the chassis manufacturer. We empirically verify that these factory-provided extrinsics are sufficiently accurate for scene-level reconstruction and navigation, without requiring additional manual refinement.

## A.2. More Experiments

**Quantitative Analysis of False-Positive Correction.** To further isolate the error-correction capability of the Evidential Experience Calibrator (EEC), we evaluate PSG-Nav on a failure-mode-specific subset of HM3D. Specifically, we first run PSG-Nav without EEC and collect all validation episodes in which the agent fails due to a false-positive stop, i.e., the agent incorrectly terminates at a visually plausible but semantically wrong object. This produces a high-ambiguity subset of 320 episodes. By construction, the success rate of the w/o EEC variant on this subset is 0.0%.

Table 5 reports the outcome breakdown after enabling EEC on the same fixed subset. PSG-Nav successfully intercepts 246 out of 320 false-positive stops, yielding a false-positive interception rate of 76.9%. Among these previously failed episodes, PSG-Nav further recovers 97 episodes and achieves a 30.3% success rate. This result directly verifies that EEC is not merely improving aggregate performance, but specifically corrects the intended failure mode by rejecting unreliable goal confirmations and forcing continued exploration.

The remaining failures also provide useful diagnostic insights. After EEC rejects false positives, the dominant failure mode shifts from premature stopping to exploration limitations: 38.1% of episodes time out because the true goal is located on another floor, while 8.4% fail because the open-vocabulary detector never fires on the real goal. These cases indicate that EEC effectively resolves false-positive confirmation errors, while the residual failures are mainly caused by low-level exploration coverage and detector recall rather than calibration.

**EEC Memory Capacity Ablation.** We further analyze the sensitivity of EEC to the memory capacity $N_{\max}$ on HM3D. Since EEC maintains positive and negative experience banks for retrieval-based confidence calibration, an important question is whether its performance depends on storing a large number of historical examples. Table 6 reports the results under different capacity limits.

The results show that EEC brings consistent gains even with a small memory bank. Increasing $N_{\max}$ from 0 to 5 improves SR from 63.5% to 65.2% and reduces the false-positive rate from 15.2% to 13.3%, confirming that a compact set of retrieved experiences is already effective for correcting unreliable goal confirmations. The default setting $N_{\max} = 10$ further improves SR to 66.1% and lowers the false-positive rate to 12.8%. Increasing the capacity to 20 only yields a marginal additional gain, suggesting that the memory benefit saturates quickly. Therefore, we choose $N_{\max} = 10$ as the default setting, which achieves a favorable trade-off between navigation performance, memory size, and retrieval overhead.

**HM3D-OVON Benchmark.** To evaluate whether PSG-Nav remains effective beyond the small-category ObjectNav setting,

*Table 5.* Quantitative analysis of EEC on the HM3D false-positive subset. The subset contains 320 episodes where PSG-Nav without EEC fails due to a false-positive stop.

| Outcome | w/o EEC | PSG-Nav |
|---|---|---|
| Success ↑ | 0.0% (0) | **30.3% (97)** |
| Fail – False Positive Stop ↓ | 100.0% (320) | **23.1% (74)** |
| Fail – Timeout: goal on other floor | 0.0% (0) | 38.1% (122) |
| Fail – Timeout: detector miss | 0.0% (0) | 8.4% (27) |
| False-Positive Interception Rate ↑ | – | **76.9% (246/320)** |

*Table 6.* Ablation study on the memory capacity $N_{\max}$ of EEC on HM3D. $N_{\max} = 0$ corresponds to removing EEC.

| Memory Capacity $N_{\max}$ | SR (%) ↑ | SPL (%) ↑ | False Positive Rate (%) ↓ |
|:---:|:---:|:---:|:---:|
| 0 (w/o EEC) | 63.5 | 31.2 | 15.2 |
| 5 | 65.2 | 31.8 | 13.3 |
| 10 (Default) | **66.1** | **32.1** | 12.8 |
| 20 | 66.3 | 32.1 | 12.7 |

*Table 7.* Comparison on the HM3D-OVON Val Seen split. PSG-Nav is training-free and uses WeDetect (Fu et al., 2025) as the open-vocabulary detector.

| Method | Type | SR (%) ↑ | SPL (%) ↑ |
|:---|:---:|:---:|:---:|
| VLFM (Yokoyama et al., 2024) | Training-free | 35.2 | 18.6 |
| TANGO (Ziliotto et al., 2025) | Training-free | 35.5 | 19.6 |
| MTU3D (Zhu et al., 2025) | Finetuned | 40.8 | 12.1 |
| CompassNav (Li et al., 2025) | Finetuned | 43.5 | 21.6 |
| JanusVLN (Zeng et al., 2025) | Finetuned | 44.9 | **31.7** |
| **PSG-Nav** | Training-free | **46.4** | 21.5 |

we further test it on the HM3D-OVON benchmark, which contains a substantially larger set of open-vocabulary goal categories. This setting is more challenging because the agent must handle a broader semantic space, more fine-grained visual ambiguity, and a larger number of possible goal descriptions.

We evaluate PSG-Nav on the HM3D-OVON Val Seen split using WeDetect as the open-vocabulary detector. As shown in Table 7, PSG-Nav achieves 46.4% SR and 21.5% SPL without any task-specific finetuning. Compared with training-free baselines, PSG-Nav outperforms VLFM and TANGO by more than 10 absolute points in SR. It also achieves competitive performance against finetuned methods, surpassing MTU3D and CompassNav in SR. These results demonstrate that the proposed probabilistic scene-graph representation and uncertainty-aware decision process generalize well to large-category open-vocabulary navigation.

The gap between PSG-Nav and training-free baselines suggests that explicitly preserving semantic uncertainty is especially beneficial when the category space becomes larger. In such settings, hard-label scene graphs are more vulnerable to irreversible perception errors, while PSG-Nav retains alternative semantic hypotheses and reasons over multiple plausible scene interpretations. We also observe that PSG-Nav prioritizes success rate over path efficiency compared with JanusVLN, which obtains a higher SPL after finetuning. This indicates that large-category navigation still leaves room for improving low-level exploration efficiency, while PSG-Nav provides strong training-free robustness under semantic ambiguity.

## A.3. Technical details

### A.3.1. DETAILS OF HYBRID LANDMARK GENERATION

We employ a hybrid strategy derived from the occupancy map to generate candidates that balance navigational safety with exploration. Primarily, we target nodes on the Generalized Voronoi Graph (GVG), which constitutes the topological skeleton of the free space. Prioritizing unvisited GVG nodes ensures the agent traverses the environment's structural core with maximized obstacle clearance. To mitigate the GVG's potential inability to cover narrow corners, we implement a fallback mechanism: when Voronoi nodes are exhausted, we target the midpoints of frontiers (boundaries between free and unknown space), forcing the agent into peripheral regions.

(1) Voronoi-based Landmark Generation: Landmarks are sampled as the sparse topological skeleton of the explored free space $\mathcal{M}_{free}$. The generation follows a rigorous filtering process:

- Generalized Voronoi Diagram (GVD): We extract boundary points from $\mathcal{M}_{free}$ and compute the GVD to find the medial axis of the environment.

- To ensure navigational reliability, candidate Voronoi vertices are pruned unless they are: i). Within the explored free

space ($\mathcal{M}_{free}$); ii). At least 20 cm away from any detected obstacles.

- Reduced Voronoi Diagram (RVD): To minimize redundancy, we simplify the graph by removing all degree-2 nodes. This results in a sparse set of landmarks consisting solely of intersections (degree $\geq 3$) and leaves (degree $= 1$), which represent intersection points and exploration frontiers respectively.

(2) Frontier-based Landmark Generation: To ensure continuous navigation in complex or degenerate environments (*e.g.*, narrow corridors or areas with sparse boundary points), we extract landmarks from frontier midpoint.

- We apply DBSCAN clustering to raw frontier points to identify continuous frontier segments.

- We select the midpoint of each sorted frontier cluster as the landmark.

- For landmarks within a $1.0m$ range of the agent, we perform an A* path-planning check on the $\mathcal{M}_{free}$. Only points with a valid, obstacle-free path to the agent's current position are retained.

A.3.2. DETAILS OF INFORMATION GAIN ESTIMATION

To quantify the exploration potential of each candidate, we evaluate the expected reduction in environmental uncertainty at each landmark $l_i \in \mathcal{L}_t$. As defined in Sec. 3.3.2, the geometric utility $I_{\text{spa}}(l_i)$ measures the expected incremental visibility of the environment. Specifically, we simulate $N_r$ rays (where $N_r = 72$) originating from $l_i$ to estimate the area of unknown regions $\mathcal{U}$ that would become observable. Each ray terminates upon encountering an obstacle or reaching the maximum sensing range $r_{\max}$. The spatial gain is calculated as the normalized area of unique unknown pixels visible from $l_i$:

$$I_{\text{spa}}(l_{i,t}) = \frac{|\mathcal{U}(l_{i,t})|}{\pi r_{\max}^2} \tag{12}$$

where $r_{\max}$ is the agent's maximum sensing range and $|\mathcal{U}(l_{i,t})|$ denotes the set of unknown regions observable from $l_{i,t}$. This implementation drives the agent to prioritize frontiers that maximize the expansion of the geometric occupancy map.

### A.4. Prompt Templates

In this section, we provide representative examples of the prompt templates used for hierarchical logical pruning and goal-selection reasoning. To reflect the algorithmic execution order, we first present the pruning prompts used to refine the 3D-PSG into a consistent multiverse, followed by the stochastic pairwise comparison prompt used for navigation decision-making.

### A.5. Group-level Logical Pruning Prompt

This prompt is used during the bottom-up construction of the multiverse to discard object configurations that exhibit internal logical conflicts or violate local spatial common sense (Sec. 3.3.1).

---

**Group-level Logical Pruning Prompt**

**Task:** You are evaluating whether object groups in indoor scenes are plausible. Below is a list of object groups, each with a room type and objects found together. For each group, determine if the combination is reasonable.
**Format:** Return your answer as a JSON array with the indices (1-indexed) of ONLY the PLAUSIBLE groups. Format: `{"plausible": [1, 3, 5, ...]}`. If all groups are plausible, return all indices. If none are plausible, return an empty array.
**Groups to evaluate:**
1. Room: unknown, Objects: cushion, cushion
2. Room: bedroom, Objects: chair, cushion, cushion
3. Room: living room, Objects: drawers, staircase
4. Room: bedroom, Objects: fitness equipment, staircase
5. Room: living room, Objects: staircase
6. Room: bedroom, Objects: fitness equipment, fitness equipment
7. Room: unknown, Objects: cushion, chair
8. Room: unknown, Objects: chair, cushion
9. Room: bedroom, Objects: chair, cushion, chair
10. Room: bedroom, Objects: chair, chair, cushion
11. Room: bedroom, Objects: stool, staircase
12. Room: living room, Objects: drawers
**Question:** Which groups represent semantically and spatially consistent configurations in an indoor environment?

- - - - - - - - - - - - - - - - - - - - - - - - - - - - - - - - - - - - - - - - - - - - - - - - - - - - - - - - - - - - - -

**LLM Response (JSON only):**

`{"plausible": [2, 7, 8, 9, 10]}`

---

### A.5.1. ROOM-LEVEL LOGICAL PRUNING PROMPT

Once group-level configurations are validated, the agent evaluates the overall room-level layout to ensure the combination of multiple groups aligns with architectural logic (Sec. 3.3.1).

---

**Room-level Logical Pruning Prompt**

**Task:** You are evaluating whether room configurations are plausible. Below is a list of room configurations, each containing multiple object groups. For each configuration, determine if the overall setup is reasonable for that room type.
**Format:** Return your answer as a JSON array with the indices (1-indexed) of ONLY the PLAUSIBLE configurations. Format: `{"plausible": [1, 2, 4, ...]}`. If all configurations are plausible, return all indices. If none are plausible, return an empty array.
**Configurations to evaluate:**
1. Room: bedroom
   Group 1: chair, cushion, cushion
   Group 2: fitness equipment, fitness equipment
2. Room: unknown
   Group 1: cushion, chair
3. Room: unknown
   Group 1: chair, cushion
4. Room: bedroom
   Group 1: chair, cushion, chair
   Group 2: fitness equipment, fitness equipment
5. Room: bedroom
   Group 1: chair, chair, cushion
   Group 2: fitness equipment, fitness equipment
**Question:** Which configurations represent logically consistent architectural layouts?

- - - - - - - - - - - - - - - - - - - - - - - - - - - - - - - - - - - - - - - - - - - - - - - - - - - - - - - - - - - - - -

**LLM Response (JSON only):**

`{"plausible": [3, 4, 5]}`

---

### A.5.2. STOCHASTIC PAIRWISE COMPARISON PROMPT

After instantiating the multiverse $\mathcal{M}$, the LLM evaluates candidate landmarks by comparing their potential to lead the agent to the goal $g$ within a specific sampled world $\mathcal{G}^{(m)}$ (Sec. 3.3.3).

---

**Stochastic Pairwise Comparison Prompt**

---

**Task:** The robot is searching for "drawers" . Compare two landmarks and decide which one is more likely to help find the goal.
**Exploration Context:**
- **Overall Progress:** Early exploration (17.8 $m^2$ mapped)
- **Room Status:**
  - Partially explored: living room (73%)
- **Goal Relevance for "drawers":**
  - living room (medium relevance, 73% explored)
  - Additional rooms may exist in unexplored areas
**Landmark A:**
- **Location:** 2.8m away, in living room
- **Surrounding:** 95% explored, far from frontiers
- **Room:** living room (100%)
- **Nearby objects:** near cushion (2.2m)
**Landmark B:**
- **Location:** 3.3m away
- **Surrounding:** 95% explored, far from frontiers
- **Room:** nearest to living room (2.4m away)
- **Nearby objects:** no detected objects nearby
**Question:** Which landmark is MORE likely to help find "drawers"?
**Consider:**
  1. Which location is more likely to have drawers based on room type and nearby objects?
  2. Which has better exploration potential (unexplored areas, near frontiers)?
  3. Which provides better information gain for finding the goal?
**Answer with ONLY "A" or "B" and a brief reason (one sentence).**

- - - - - - - - - - - - - - - - - - - - - - - - - - - - - - - - - - - - - - - - - - - - - - - - - - - - - - - - - - - - - -

**LLM Response:** *A. The living room is more likely to contain drawers based on room type and the presence of nearby objects like a cushion, which suggests a more domestic setting.*

---

## A.6. Hyperparameter Settings

Table 8 summarizes the key hyperparameters used in PSG-Nav.

*Table 8.* Detailed hyperparameters for PSG-Nav.

| Category | Parameter | Value |
|----------|-----------|-------|
| **3D-PSG** | Object category count vector $n_{i,t}$ | $\mathbb{R}^{|C|}$ |
| | Group threshold $K_g$ | 16 |
| | Room threshold $K_r$ | 16 |
| **Multiverse Decision** | Information gain weight $\alpha$ | 1.0 |
| | Decision fusion weight $\beta$ | 0.5 |
| | Candidate filter threshold $\tau$ | 0.1 |
| | Multiverse sampling count $K$ | 3 |
| **RAG Verifier** | Confidence threshold $\delta$ | 0.61 |
| | Negative penalty weight $\gamma$ | 2.0 |
| | Memory capacity $N_{max}$ | 10 |
| | Similarity weights $w_1, w_2$ | 0.5, 0.5 |
| **Model Backbone** | LLM Reasoning Engine | Qwen2.5-7B-Instruct |
| | Open-vocabulary Detector | GLIP |
| | Instance Segmentor | Grounded-SAM |

## A.7. Failure Case Analysis

To provide a deeper understanding of the system's limitations, we conduct a comprehensive analysis of the failure episodes of PSG-Nav on the HM3D datasets. The failure cases can be primarily categorized into three types:

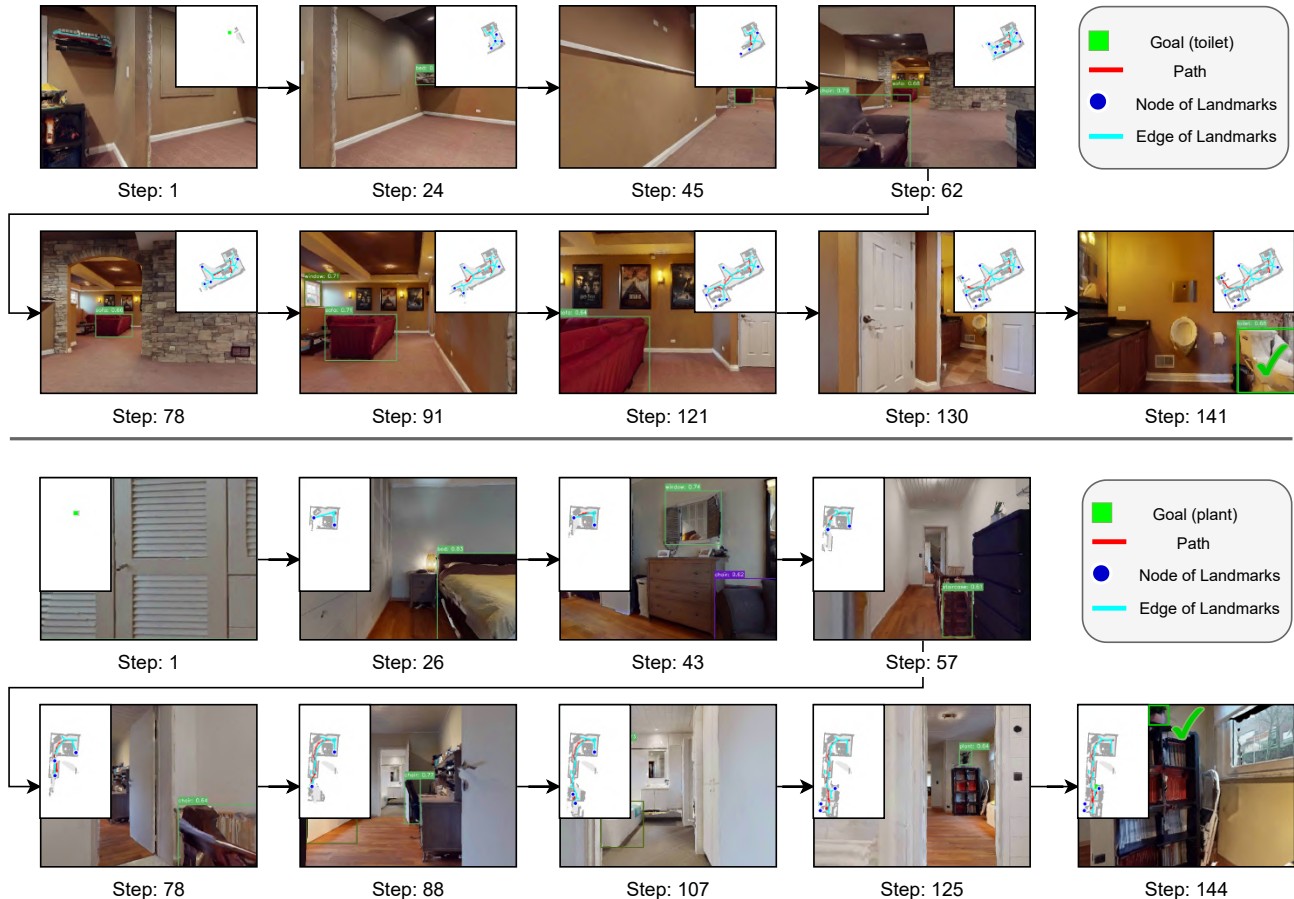

*Figure 5.* Visualization of the navigation process of PSG-Nav on HM3D.

(1) Perception-Driven False Positives (37.5%): This is a primary failure mode where the agent incorrectly identifies a non-target object as the goal and prematurely executes the STOP action. This often occurs when the open-vocabulary detector (*e.g.*, GLIP) generates a high raw detection score $S_{det}$ for visually similar objects. If the RAG Verifier lacks sufficiently distinct historical experiences in its memory banks ($\mathcal{B}^+, \mathcal{B}^-$) to generate a strong negative calibration signal $\Delta S$, the final calibrated score $S_{final}$ remains above the termination threshold, leading to a false success declaration.

(2) Geometric Constraints (40.2%): A significant portion of failures, particularly in multi-floor environments and missing meshes, stems from our reliance on a 2D occupancy map for exploration. When the agent encounters stairs leading to a goal on a different elevation, the 2D projection often misidentifies the vertical rise of the steps as a non-traversable obstacle. This causes a navigation deadlock where the agent is unable to plan a valid path to the target floor, resulting in aimless wandering or an episode timeout.

(3) Missing Goal (22.3%): In these cases, the agent successfully navigates to the vicinity of the goal object, but the perception system fails to recognize it. This failure is often caused by extreme viewpoint variations, heavy occlusions, or poor lighting conditions that cause the detection confidence to fall below the activation threshold.

### A.8. Visualization

As shown in figure 5, 6, and 7, we provide qualitative visualizations on HM3D, MP3D, and HSSD to highlight the navigation process of PSG-Nav.

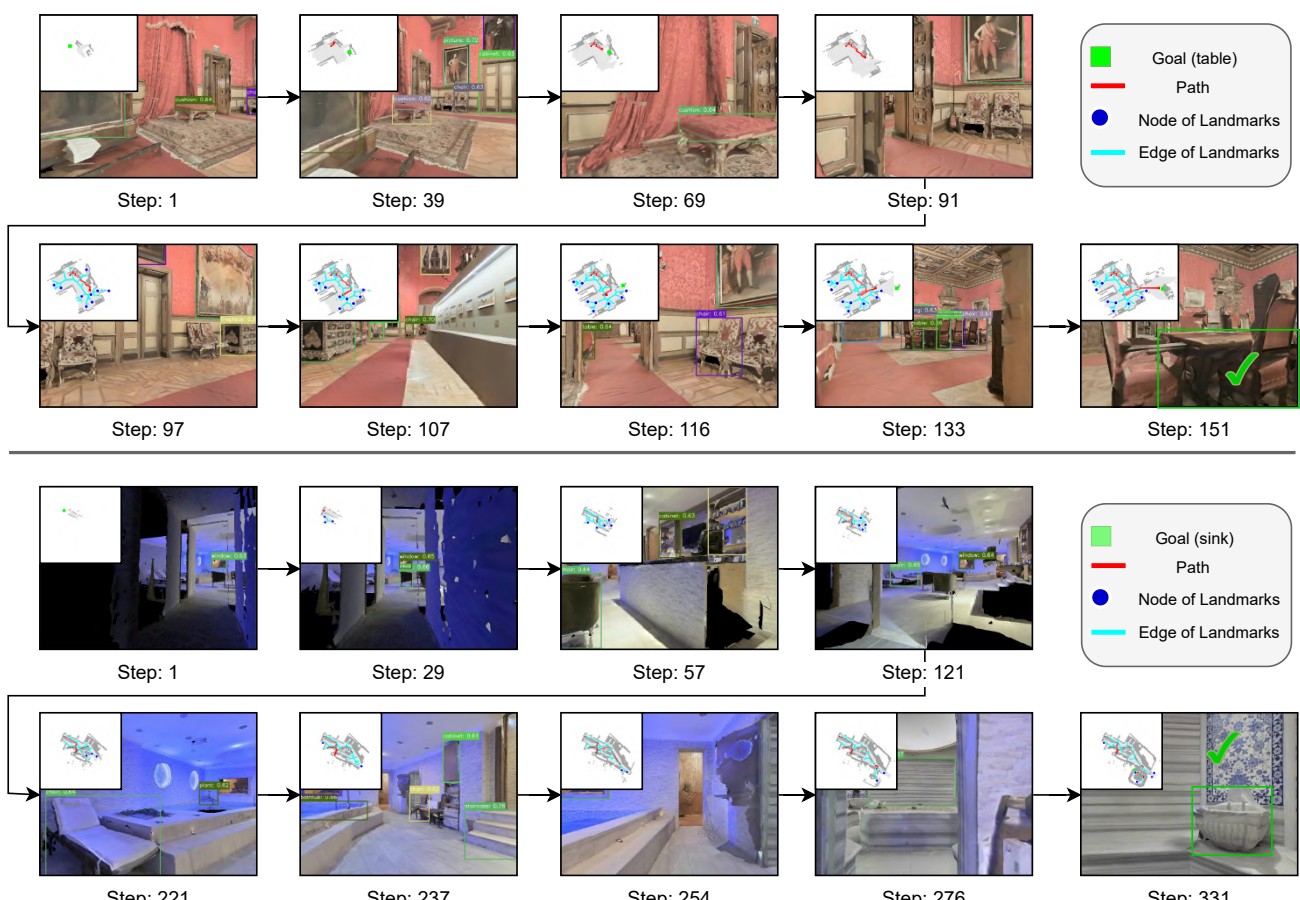

*Figure 6.* Visualization of the navigation process of PSG-Nav on MP3D.

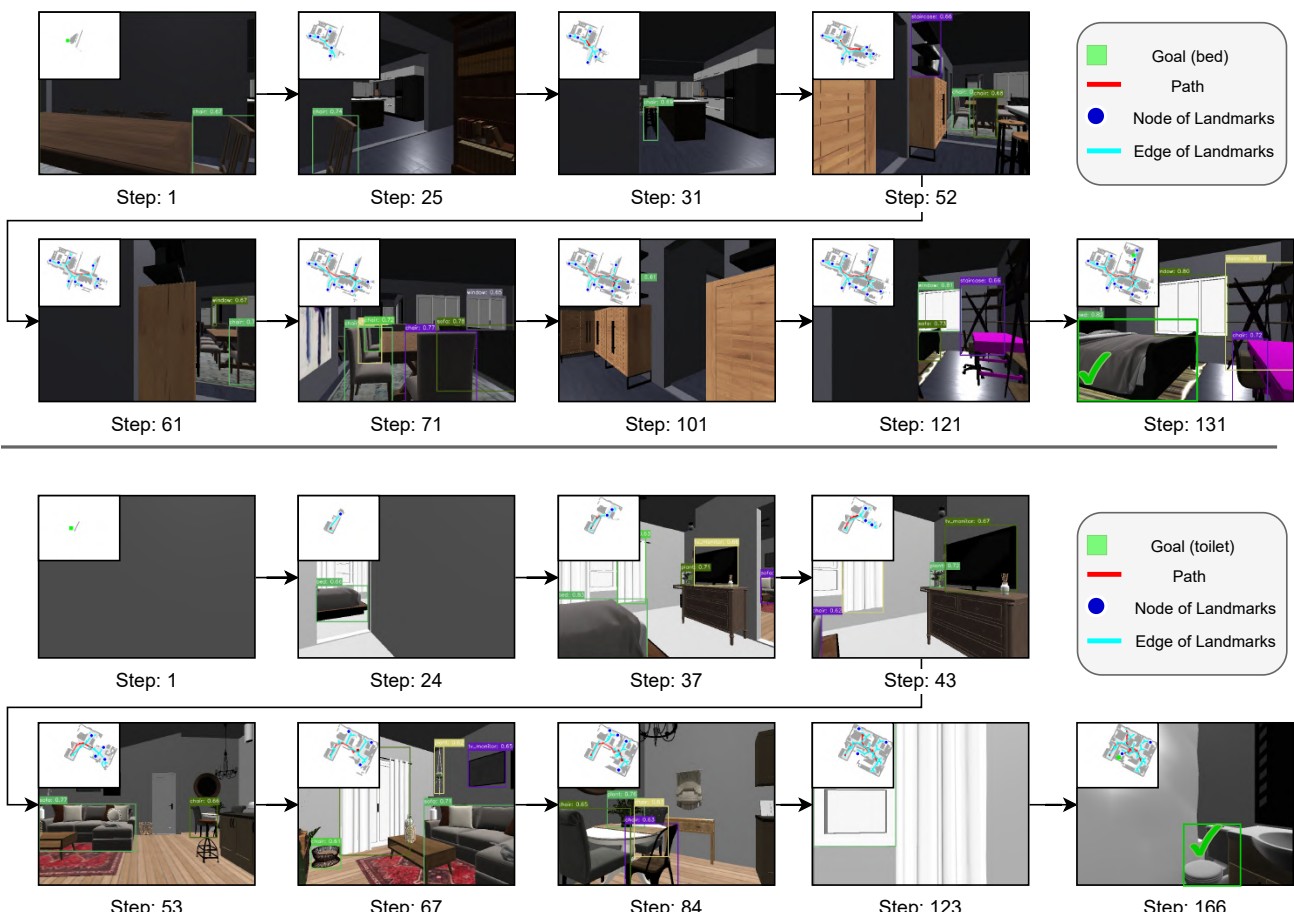

*Figure 7.* Visualization of the navigation process of PSG-Nav on HSSD.

---

**Algorithm 1** Overall PSG-Nav Pipeline

---

**Require:** Environment $\mathcal{E}$, Target Object $O_{\text{target}}$, LLM Oracle $f_{\text{LLM}}$, Memory Banks $\mathcal{B}^+, \mathcal{B}^-$
**Ensure:** Navigation execution success or failure
 1: Initialize 3D-PSG nodes $\mathcal{N} \leftarrow \emptyset$
 2: **while** Max Steps not reached **do**
 3:     *// Phase 1: Perception & 3D-PSG Update*
 4:     Obtain observation $O_t$ and update object beliefs $n_{i,t}$ via vote accumulation (Eq. 1)
 5:     *// Phase 2: Goal Confirmation via EEC (Sec 3.4)*
 6:     **if** Candidate object $o_c$ identified with initial score $S_{\text{det}}$ **then**
 7:         Query memory banks to compute $S_{\text{pos}}$ and $S_{\text{neg}}$ via composite similarity (Eq. 9, 10)
 8:         Calculate calibration margin $\Delta S = S_{\text{pos}} - \gamma \cdot S_{\text{neg}}$
 9:         Calibrate final confidence $S_{\text{final}} = S_{\text{det}} + \Delta S$
10:         **if** $S_{\text{final}} > \delta$ **then**
11:             Execute STOP action
12:             Update $\mathcal{B}^+, \mathcal{B}^-$ using redundancy-aware pruning (Eq. 11)
13:             **return Success**
14:         **end if**
15:     **end if**
16:     *// Phase 3: Multiverse Decision & Exploration (Sec 3.3)*
17:     Generate candidate landmarks $\mathcal{L}_t$ from GVG and geometric frontiers
18:     Compute epistemic utility $U_{\text{gain}}(l_{i,t}) = \alpha I_{\text{spa}} + I_{\text{sem}}$ (Eq. 4, 5)
19:     Filter high-potential landmarks $\mathcal{L}'_t = \{l \in \mathcal{L}_t \mid U_{\text{gain}}(l) \geq \tau\}$ (Eq. 6)
20:     Sample discrete multiverse $\mathcal{M}$ from 3D-PSG, applying logical filter $f_{\text{LLM}}(s)$ (Eq. 3)
21:     Perform stochastic pairwise comparisons $\mathbb{I}$ across $\mathcal{M}$ using LLM (Eq. 7)
22:     Compute marginalized preference score $S(l_{i,t})$ (Eq. 8)
23:     Select optimal landmark $l^* = \arg\max(S(l_{i,t}) + \beta U_{\text{gain}}(l_{i,t}))$
24:     *// Phase 4: Action Execution*
25:     Navigate toward $l^*$ using local physical controller
26: **end while**
27: **return Failure**

