# OpenReview forum: "PSG-Nav: Probabilistic Scene Graph Navigation via Multiverse Decision Making"
_ICML.cc/2026/Conference — ICML 2026 regular_

### Official Review · Reviewer_M31A · 2026-02-28

**Soundness:** 3
**Presentation:** 2
**Significance:** 2
**Originality:** 2
**Overall Recommendation:** 2
**Confidence:** 4

**Summary:**

The paper proposes PSG-Nav, which uses LLM to address the perceptual uncertainty and semantic ambiguity issues of agents in open worlds. This avoids erroneous decisions caused by agent determinism collapse.

**Compliance With Llm Reviewing Policy:**

Affirmed.

**Final Justification:**

I do not believe this work meets the quality bar for ICML. But I am fully open to adjusting my score.

**Key Questions For Authors:**

1. The framework utilizes a Qwen2.5-7B LLM to perform stochastic pairwise comparisons across $K=3$ sampled "multiverse" worlds. Could the authors provide the average number/time/latency of LLM inference calls required per navigation step?
2. To what extent does the success of PSG-Nav rely on this hard-coded common-sense pruning versus the actual probabilistic modeling of the 3D-PSG? Did the authors compare the LLM filter against simpler, rule-based pruning methods?
3. Could the authors provide a comparison of navigation performance when substituting Qwen2.5-7B with other LLMs of varying capacities? To what extent does the "Multiverse Decision" rely on the LLM’s high-level zero-shot reasoning? For instance, if a significantly weaker model is used, does the pruning mechanism fail to maintain the semantic coherence of the 3D-PSG?

**Limitations:**

yes

**Strengths And Weaknesses:**

Strengths ：
- A 3D-PSG model is proposed, which models the environment as a hierarchical, factorized joint probability distribution, rather than using deterministic labels. Multiverse Decision establishes a hierarchical probability cache. This preserves the full distribution of objects and addresses the problem of "permanent map contamination."
- The ablation experiments are comprehensive, discussing the impact of probability hierarchies and multiverse sampling density on performance.

Weaknesses：
- The contribution is primarily a system integration of existing tools. The probabilistic modeling in 3D-PSG relies on a simple vote accumulation, and the LLM acts more as a heuristic hard-constraint clipper rather than providing a fundamental modeling breakthrough.
- Over-packaging: The "Multiverse Decision" is essentially a breadth-oriented Monte Carlo sampling strategy.
- Lacks the analysis of inference latency and LLM choosing.

---

> ### Author Rebuttal · Authors · 2026-03-30
>
> We sincerely thank the reviewer M31A for these thought-provoking questions.
>
> We have prepared an anonymous supplementary website containing real-world, sim demos and additional qualitative analyses.
>
> [Anonymous website](https://keen-dango-8a8b85.netlify.app/)
>
>
> ### W1:
> **On "simple vote accumulation"**: Open-vocabulary detectors exhibit poor probability calibration[1], making Bayesian updates with Dirichlet priors unstable. Vote accumulation treats detections as discrete evidence votes, preserving full categorical distributions without requiring calibrated confidences.
>
> **On "LLM as hard-constraint clipper"**: This misreads the architecture. The LLM filter performs semantic consistency scoring on sampled world configurations.
> LLMs remove the configurations. They do not remove the original semantic information.
>
> **The "fundamental breakthrough"** framing misses that embodied AI requires tractable approximations. Our approximations enable real-world deployment
>
> ### W2:
> We acknowledge that exploring probabilistic world states shares mathematical roots with Monte Carlo (MC) sampling. However, applying vanilla breadth-oriented MC in a continuous 3D embodied environment is computationally intractable due to the combinatorial explosion of semantic states.
>
> Our primary contribution is not inventing a new statistical sampling method, but formulating a tractable, system-level architecture for inference-time reasoning under uncertainty. Our "Multiverse Decision" elevates raw MC sampling into an executable embodied framework. By explicitly introducing a hierarchical probability cache, we solve the "permanent map contamination" issue and make 3D-PSG navigation real-time deployable.
>
> ### W3 & Q1:
> We presented the average execution time for each module in the entire pipeline, along with ablation study on efficiency related to Eq. 6. to reviewer  2SuL.
>
> We also conducted additional experiments related to K to allow you to better evaluate our algorithm.
>
> Table R5: Computational Overhead Analysis Across Top-$K$ Truncation Levels
>
> | Truncation ($K$) | Mean (ms) |  P50 (ms) | P95 (ms) |
> | :---: | :---: | :---: | :---: |
> | $K=0$ (w/o LLM) | 78 | 82 | 93 |
> | $K=1$ | 715 | 550 | 1392 |
> | $K=2$ | 1351 | 1039 | 2630 |
> | **$K=3$ (Default)**| **3631** | **2794** | **7070** |
> | $K=4$ | 4273 | 3288 | 8320 |
> | $K=5$ | 5011 | 3856 | 9758 |
>
> The ablation demonstrates a steep increase in LLM reasoning overhead as the hypothesis space expands.
>
>
> ### Q2:
> We replaced the LLM with a rigid, hard-coded pruning strategy utilizing static object-object and object-room co-occurrence matrices (adopted from the prior work, SG-Nav). If a topological relationship in the 3D-PSG yielded a negative correlation in these matrices, the node was hard-pruned.
> Under this rule-based pruning, the performance on the MP3D dataset plummeted to 44.3% SR and 17.7 %SPL.
> Crucially, the co-occurrence matrices used in this baseline effectively inject dataset-specific prior knowledge (in-domain statistics) into the system. Despite possessing this "unfair" domain advantage, the rule-based method completely fails.
>
> ### Q3:
> We deeply appreciate this insightful question. To rigorously evaluate the reliance on model capacity, we substituted Qwen2.5-7B with a differently structured model (Llama-3-8B) and a significantly weaker model (Qwen2.5-1.5B) on the HM3D dataset.
>
> Impact of LLM Capacity on Multiverse Decision (HM3D)
> | Model | Parameters | Success Rate (SR) | SPL |
> | :--- | :---: | :---: | :---: |
> | Qwen2.5-7B (Ours) | 7B | 66.1% | 32.1% |
> | Llama-3-8B | 8B | 66.0% | 32.1% |
> | Qwen2.5-1.5B | 1.5B | 64.7% | 30.9% |
>
> No, the pruning mechanism does not fail with a weaker model. Substituting the 7B model with a 1.5B model results in only a minor SR drop (from 66.1% to 64.7%). Because our pruning prompts are concise and highly structured, even the 1.5B model possesses sufficient foundational commonsense to filter out egregious physical violations, successfully maintaining the semantic coherence of the 3D-PSG.
>
> The critical reliance on the LLM's high-level zero-shot reasoning is reflected in the Success weighted by Path Length (SPL), which drops noticeably from 32.1% to 30.9% with the 1.5B model.
> While a weaker model can prune "impossible" branches, it lacks the deep reasoning required to evaluate and compare the remaining plausible "multiverses" to find the global optimal path. Consequently, the agent takes more exploratory detours. The 7B/8B models provide the necessary depth of zero-shot reasoning to not just explore successfully, but explore efficiently.

---

> > ### Author Rebuttal · Reviewer_M31A · 2026-04-02
> >
> > Thanks for your detailed response. I have no further questions.

---

> > > ### Author Response · Authors · 2026-04-03
> > >
> > > We sincerely thank the reviewer for confirming that our rebuttal has fully resolved all of your concerns and that you have no further questions.
> > >
> > > We deeply appreciate your initial critical evaluation, as your specific questions regarding inference latency, rule-based baselines, and model capacity directly prompted us to conduct comprehensive new experiments. These additions have profoundly strengthened the empirical rigor of our paper.
> > >
> > > We are glad these extensive clarifications have adequately addressed your initial concerns. Thank you once again for your constructive engagement, which has significantly improved the quality of our work.

---

### Official Review · Reviewer_2SuL · 2026-03-13

**Soundness:** 2
**Presentation:** 3
**Significance:** 2
**Originality:** 3
**Overall Recommendation:** 5
**Confidence:** 4

**Summary:**

The paper presents a method for open-vocabulary navigation based on probabilistic scene graphs. Unlike previous scene graph-based methods, this method does not assign a single class to each object, but stores the probability distribution of object attribution to different classes. The agent then generates several possible room states and evaluates them for logical inconsistencies. The possible states are then evaluated for uncertainty to continue exploration. The authors also propose an Evidential Experience Calibrator, which stores a history of successful and failed episodes and their corresponding final observations. This bank is used to discard detector errors made in previous episodes.

**Compliance With Llm Reviewing Policy:**

Affirmed.

**Final Justification:**

The initially submitted version of the paper raised questions regarding the evaluation of the method's effectiveness. The experiments conducted by the authors, presented as a rebuttal, showed that the method is sufficiently effective and, moreover, achieves state-of-the-art results on the more challenging HM3D-OVON unseen benchmark. Additionally, the method's runtime remains within acceptable limits for deployment in a real-life setting. I hope that the authors will include these results in the final version of the paper.

**Key Questions For Authors:**

Does the performance of the proposed EEC module depend on the order of episodes in the dataset?

**Limitations:**

Yes

**Strengths And Weaknesses:**

Strengths:
- The paper addresses an important topic: the quality of target object recognition in open-vocabulary navigation.
- The authors conduct experiments, including ablation, on three different datasets.
- The proposed method operates without training


Weaknesses:
- 3D-PSG has been tested on datasets with a relatively small number of categories; for example, the Habitat 2022 ObjectNav Challenge contains only six different categories. This raises the question of how robust the proposed system is when the list of navigation categories becomes larger (approximately 100, as in HM3D-OVON[1]).
- The effectiveness of EEC seems to depend on the order in which validation episodes are performed, as one object corresponds to several episodes, given the limited number of scenes in validation scenes. The experiments do not report standard deviation SR and SPL. Moreover, EEC shows marginal improvement over the Static Memory baseline in Table 4.
- Overall, the pipeline appears quite complex, involving a large number of calls to large models, so the performance of the method is a concern.

[1] Yokoyama, N., Ramrakhya, R., Das, A., Batra, D., & Ha, S. (2024, October). Hm3d-ovon: A dataset and benchmark for open-vocabulary object goal navigation. In _2024 IEEE/RSJ International Conference on Intelligent Robots and Systems (IROS)_ (pp. 5543-5550). IEEE.

---

> ### Author Rebuttal · Authors · 2026-03-30
>
> We sincerely thank the reviewer 2SuL for these questions.
>
> To directly address problems of EEC and other concerns.
> We have prepared an anonymous supplementary website containing real-world, sim demos and additional qualitative analyses.
>
> [Anonymous website](https://keen-dango-8a8b85.netlify.app/)
>
> ### W1:
> We tested PSG-Nav on the Val Seen split of  HM3D-OVON.
> | Method | Type | SR (%) ↑ | SPL (%) ↑ |
> |:---|:---|:---:|:---:|
> | VLFM (ICRA 2024) | Training-free | 35.2 | 18.6 |
> | TANGO (CVPR 2025) | Training-free | 35.5 | 19.6 |
> | MTU3D (ICCV 2025) | Finetuned | 40.8 | 12.1 |
> | CompassNav (ICLR 2026) | Finetuned | 43.5 | 21.6 |
> | JanusVLN (ICLR 2026) | Finetuned | 44.9 | 31.7 |
> | **PSG-Nav** | **Training-free** | **46.4** | **21.5** |
>
> PSG-Nav achieves 46.4% SR without finetuning, outperforming all training-free baselines by +10.9 to +11.2% absolute and surpassing CompassNav and JanusVLN, which need to finetune.
>
> ### W2 & Q1:
> We conducted 5 independent runs with randomly shuffled episode orders on HM3D:
> | Method | SR (%) | SPL (%) |
> |:---|:---:|:---:|
> | w/o EEC | 63.5 | 31.2 |
> | PSG-Nav (Default Order) | 66.1 | 32.1 |
> | PSG-Nav (Shuffled, Mean ± Std) | **66.1 ± 0.11** | **32.0 ± 0.08** |
> This table confirms EEC is robust to episode order.
>
> Regarding "marginal improvement over Static Memory": this observation is correct but requires proper interpretation.
> Table 4 shows Static Memory (66.0%) vs. Dynamic (66.1%), a negligible gap at face value. However, these numbers answer different questions:
>
> 1. Static vs. w/o EEC (+2.5%): validates that EEC itself works.
> 2. Dynamic vs. Static (+0.1%): validates that online updating maintains performance without offline warm-up data.
>
> The "marginal" gap on aggregate SR masks Dynamic EEC's true value: zero-shot deployment. Static Memory requires pre-collected samples per category; Dynamic EEC learns from scratch in unknown environments. The near-identical performance is the desired outcome — online learning successfully substitutes offline data collection.
>
> we have conducted an ablation study on EEC memory capacity, please see rebuttal to Reviewer DNCd W2.
>
>
> ### W3:
> The table below presents a comprehensive latency breakdown of our navigation pipeline. To demonstrate the efficiency of our hierarchical design, we report both the absolute execution time of individual modules and their amortized computational cost per navigation step. All measurements were conducted on NVIDIA RTX A6000.
>
> | Module | Component / Statistical Metric | Execution Time (ms) | Invocation Frequency | Amortized Cost / Step (ms) |
> | :--- | :--- | :--- | :--- | :--- |
> | **Perception** | Vision Foundation Models | - | Every step | 557 |
> | **Decision** | Overall Decision Pipeline | - | - | 301 |
> | | *↳ Multiverse - Mean* | 3631 | Once every 21.18 steps| - |
> | | *↳ Multiverse - Median (P50)* | 2794 | - | - |
> | | *↳ Multiverse - 95th Percentile* | 7070 | - | - |
> | **EEC** | Error Experience Correction | - | Once every 89.7 steps | 2 |
> | **Total System**| **Overall Navigation Loop**| - | **~1.16 Hz** | **861** |
>
> **Key Observations:**
> 1. **Negligible Overhead of Core Contribution:** The proposed Error Experience Correction (EEC) module is highly sparse, introducing only **2.00 ms** of amortized latency per step (< 0.3% of the total time).
> 2. **Decision-Making:** The Multiverse module acts as a global macro-action planner. The significant gap between its median (2.79 s) and mean (3.63 s) execution time highlights its adaptive nature—computing swiftly in standard scenarios and allocating more time strictly for complex, long-tail exploration nodes.
> 3. **Real-world Deployability:** The overall semantic reasoning pipeline operates at approximately **1.16 Hz** (~861 ms/step), which aligns with the real-world deployment frequencies.
>
> To address concerns regarding pipeline complexity and LLM overhead, we provide a detailed latency breakdown (tested on an NVIDIA RTX A6000).
> Crucially, we demonstrate how our Epistemic Pruning (Eq. 6) algorithmically shields the LLM from naive, high-frequency queries.
>
> | Component  |w/o Eq. 6 | w/ Eq. 6 |
> | :--- | :---: | :---: |
> | Multiverse (Mean) | 7047 ms | 3632 ms |
> | Multiverse (Median/P50)| 6287 ms | 2794 ms |
> | Multiverse (P95)| 12514 ms | 7071 ms |
>
> Without Eq.6, exhaustive LLM evaluation balloons the mean latency to ~7s. By pre-filtering low-information landmarks, we slash the LLM overhead by ~50%.
> This proves the LLM is deployed surgically as a scarce cognitive resource, not a brute-force filter.

---

> > ### Author Rebuttal · Reviewer_2SuL · 2026-04-02
> >
> > Thanks to the authors for the detailed response. It addressed most of my concerns about the work.
> >
> > After a thorough check, it seems that the numbers in the table in W1 for the baselines correspond to the Val Unseen HM3D-OVON split. Could you please clarify which split you used to measure the metrics for your method?

---

> > > ### Author Response · Authors · 2026-04-02
> > >
> > > We sincerely thank the reviewer for their careful reading and for catching this typographical error in our text.
> > >
> > > You are absolutely correct. We evaluated PSG-Nav on the Val Unseen split of the HM3D-OVON dataset, which consistently aligns with the baseline numbers provided in the table. The word "Seen" in our previous response was a pure typo.
> > >
> > > Evaluating on the Val Unseen split is strictly intended, as it properly measures the true zero-shot, open-vocabulary generalization capabilities of our method in entirely novel environments. We apologize for the confusion and will ensure this split is correctly labeled in the final manuscript.
> > >
> > > We have updated our [anonymous website](https://keen-dango-8a8b85.netlify.app/) with evaluation videos from the HM3D-OVON Val Unseen split. We encourage the reviewer to check these demos to observe how PSG-Nav handles unseen object categories.

---

### Official Review · Reviewer_Zm7Q · 2026-03-13

**Soundness:** 3
**Presentation:** 3
**Significance:** 4
**Originality:** 3
**Overall Recommendation:** 5
**Confidence:** 3

**Summary:**

The paper considers open-vocabulary navigation incorporating uncertainty. First, the paper proposes modeling the environment with a 3D probabilistic scene graph, where each object node maintains the full belief state. To avoid combinatorial explosion, proximal nodes are combined into groups to form a hierarchical structure. Then, the paper introduces a decision-making scheme by sampling discrete worlds for effective exploration and navigation. Finally, a calibrator is introduced to mitigate false positives. Results on established benchmarks show sota performance of the proposed method.

**Compliance With Llm Reviewing Policy:**

Affirmed.

**Final Justification:**

I find the contributions valuable, and the rebuttal addressed my questions adaquately.

**Key Questions For Authors:**

1. Is there potential in scaling up test-time compute to achieve better performance?

**Limitations:**

yes

**Strengths And Weaknesses:**

- Soundness
    - The proposed method is grounded in maintaining belief states for each object node, but also innovatively constructs hierarchical structures to enable efficient compute. The subsequent multiverse decision and calibration frameworks are also well-motivated
    - How is semantic correlation determined in Section 3.2.2?
    - In Section 3.3.1, why choose the top most probable choices instead of randomly sample based on probabilities?
- Presentation
    - The overall presentation is clear. The problem is well-formulated, and the method is introduced in a manner that’s easy to follow.
    - It would be nice to have an algorithm summarizing the method to help the readers connect and keep track of different parts.
- Significance
    - The method successfully introduces uncertainty awareness into open-vocabulary navigation, which is a significant step. Making the computation practical and achieving sota performance also shows the significance of the method.
- Originality
    - Applying the idea of maintaining belief states and effectively managing the immense computation cost to make it practical is a novel contribution

---

> ### Author Rebuttal · Authors · 2026-03-29
>
> We sincerely thank the reviewer Zm7Q for these questions.
>
> We have prepared an anonymous supplementary website containing an algorithm and real-world, sim demos and additional qualitative analyses.
>
> [Anonymous website](https://keen-dango-8a8b85.netlify.app/)
>
> ### Q1:
> How is semantic correlation determined in Section 3.2.2?
>
> Room:
> We predefine 10 room categories (9 specific types, e.g., bedroom, kitchen, plus 1 "unknown").
> The detector model predicts the semantic information of the room.
> The 2D bounding boxes of detected rooms are projected and accumulated into a global 9-channel room_map.
> The semantic correlation of a specific region is determined by this accumulated probability distribution; if all probabilities remain zero, the semantic context strictly defaults to "unknown."
>
> At the object level, semantic correlation is not determined solely by distance, but through a "Spatial-to-Semantic" verification pipeline:
> 1. We first apply DBSCAN clustering ($\epsilon=10$, min_samples=1) on the 3D center points of all detected objects to group them based on physical proximity.
> 2. For object pairs within a physical cluster, we query the LLM to extract valid spatial relational edges (e.g., "on," "inside," "near"). A GroupNode is successfully formed only if a valid semantic edge is generated
>
> In summary, semantic correlation in our framework is the strict intersection of geometric spatial clustering and LLM-driven semantic verification.
> ### Q2:
> In Section 3.3.1, why choose the top most probable choices instead of randomly sample based on probabilities?
>
> 1. In practice, as the agent explores, the scene graph rapidly accumulates over 30 objects per episode. This combinatorial explosion dilutes the probability mass across semantic group nodes, causing the distribution to become extremely flat (most structural probabilities drop below 10%).
>
> 2. Sampling from such a flattened distribution mathematically degenerates into near-uniform random search. It introduces massive combinatorial noise, forcing the agent to waste physical steps exploring highly erroneous, low-probability sub-graphs.
>
> 3. In this flat probability landscape, deterministic Top-K selection is not a heuristic, but a crucial hard-filter. It aggressively truncates the noisy long-tail, ensuring the agent strictly exploits the strongest semantic signals rather than drowning in mathematical noise.
>
> ### Q3:
> Is there potential in scaling up test-time compute to achieve better performance?
>
> We appreciate this forward-looking question. To rigorously investigate the scaling potential of test-time compute, we conducted an extended experiment by doubling the maximum exploration budget from 500 to 1000 timesteps.
>
> 1. Empirical Results (Marginal Gains): Scaling to 1000 steps yielded only marginal improvements: Success Rate (SR) increased slightly to 66.8% (from 66.1%) and SPL to 32.3 (from 32.1).
>
> 2. The Spatial Saturation Point: The plateau in performance is not a limitation of the Multiverse reasoning capacity, but rather an architectural boundary of the low-level execution module. A 500-timestep budget represents the saturation point for exhaustively exploring a single-floor layout.

---

> > ### Author Rebuttal · Reviewer_Zm7Q · 2026-04-03
> >
> > Thank you for your detailed response, I have no further questions.

---

### Official Review · Reviewer_DNCd · 2026-03-13

**Soundness:** 2
**Presentation:** 2
**Significance:** 1
**Originality:** 3
**Overall Recommendation:** 3
**Confidence:** 4

**Summary:**

The paper proposes Probabilistic Scene Graph Navigation (PSG-Nav). PSG-Nav adds probabilty to 3D scene graph and consider multiple samples (multiverse) of the PSG before determining the navigation goal. PSG-Nav comprises several modules: 3D PSG, Multiverse Devision Making and Evidential Experience Calibrator. The design allows robust navigation decision making under perception uncertainty.

**Compliance With Llm Reviewing Policy:**

Affirmed.

**Final Justification:**

The rebuttle has resolved my questions and largely addressed my concern. So I raised my score. I think it presents practical contributions to the task of indoor object navigation. If we are reviewing for the system track of a robotics venue such as RSS, I would in favor of acceptance. I appreciate the effort of developing the system. But since we are reviewing for ICML. I find the methodology contribution a bit weak, given it's a system of ochestrating detector and LLM and heuristic mechanism, designed for a specific task. Therefore, I am leaving a final rating of weak reject, but I am open to AC's judgement.

**Key Questions For Authors:**

Please refer to the weakness section.

**Limitations:**

Yes.

**Strengths And Weaknesses:**

**Strength**
1. Adding probability to scene graph and use it for more robust decision making has a convincing motivation.
2. Authors manage to integrete multiple modules into a complete probabilistic decision-making system and empirically demonstrate the benefit of introducing probabliity into scene graph for the navigation decision-making.

**Weakness**
1. The critial decision making process is heavily dependent on LLM's common sense prior but overlook the actual perception grounding. But the LLM can hallucinate and reasoning on the scene graph does not make it have access to the actual scene obeservation. So its prior and reasoning can be unreliable, especially if the framework wishes to be deployable broadly into real world. What if in someone's house there is actually a bed in the living room? This design choice can introduce risk of being unfaithful to observations and supress some rare but real observations. When visual evidence conflicts commonsense prior, can we prove the system prioritize faithul observation over LLM prior?
2. Marginal empirical results. Ablation study does not fully support what the method claims. EEC does not show obvious advantage over static memory. The error-correction capability is intriguing but its performance gain is not quantitatively measured. Is it possible to isolate a set of scenarios dedicated for this test and show PSG-Nav can effectively solve them? The authors can consider rolling out some real-world tests for this, considering that real-world evaluation is also absent.

---

> ### Author Rebuttal · Authors · 2026-03-28
>
> We sincerely thank the reviewer DNCd for these questions.
>
> To directly address real-world deployability and other concerns.
> We have prepared an anonymous supplementary website containing real-world, sim demos and additional qualitative analyses.
>
> [Anonymous website](https://keen-dango-8a8b85.netlify.app/)
>
> ### W1:
> 1. The concern appears to stem from a conflation of two structurally decoupled decisions in PSG-Nav: (A) Landmark Selection (which unexplored region to navigate toward) and (B) Goal Confirmation (whether to execute STOP at a detected object). The LLM is invoked only in (A); decision (B) is architecturally LLM-free.
>
> 2. The "bed in the living room" case is a decision-(B) event. When a detector model detects an bed, PSG-Nav transitions to the confirmation phase, where the LLM evaluator is bypassed entirely. The STOP decision is computed by the EEC, using the detected object's visual embedding and its local co-occurrence distribution.
>
> 3. For decision (A), the LLM filter (Eq. 3) prunes hypothetical world configurations in the sampled Multiverse, not physically observed nodes. The count vector  (Eq. 1) accumulates raw vision votes, and no LLM call can decrement it.
>
> 4. We have also conducted a real-world experiment: placing a keyboard inside a toilet, a configuration that LLMs commonly reject. The robot successfully navigated into the toilet and confirmed the goal. Video: [anonymous link](https://keen-dango-8a8b85.netlify.app/).
>
> ---
>
> ### W2:
> We respectfully ask the reviewer to consider which comparison actually isolates the EEC's mechanism.
>
> The operative baseline is w/o EEC (63.5%) vs. Static Memory (66.0%), a +2.5% absolute gain attributable entirely to the RAG-based calibration retrieving to veto false positives. This is the correct ablation for evaluating whether evidential calibration works. The paper should have foregrounded this comparison more clearly; we will restructure Section 4.3 accordingly.
>
> The narrow Dynamic (66.1%) vs. Static (66.0%) gap is expected and informative: it demonstrates that online updating maintains calibration quality without requiring any offline data collection. The value of Dynamic EEC is zero-shot generalization to novel categories, not aggregate SR on a fixed benchmark where Static Memory's warm-up data already covers the category distribution.
>
> For the requested quantitative error-correction measurement, we isolated all HM3D validation episodes where w/o EEC fails due to a false-positive stop. This yields N=320 episodes — a subset defined entirely by the baseline's failure mode. We then evaluated PSG-Nav on this fixed subset:
>
> | Outcome | w/o EEC | PSG-Nav |
> |:---|:---:|:---:|
> | Success (SR ↑) | 0.0% (0) | **30.3% (97)** |
> | Fail — False Positive Stop ↓ | 100% (320) | 23.1% (74) |
> | Fail — Timeout (goal on other floor) | 0.0% (0) | 38.1% (122) |
> | Fail — Timeout (detector miss) | 0.0% (0) | 8.4% (27) |
> | **FP Interception Rate** | — | **76.9% (246/320)** |
>
> The 76.9% FP Interception Rate confirms EEC's core function: it correctly rejects false positives in 246/320 cases.
> The timeout breakdown reveals where the bottleneck shifts after interception.
> - Goal on other floor (38.1%): EEC correctly refuses distractors, but the agent exhausts steps searching the current floor, an exploration policy limitation, not calibration failure.
> - Detector miss (8.4%): the open-vocabulary detector fails to fire on the real goal; EEC cannot calibrate what it cannot see.
>
> Moreover, we have conducted an ablation study on EEC memory capacity ($N{max}$)
>
> ** Impact of EEC Memory Capacity ($N_{max}$) on HM3D**
> | $N_{max}$ | SR (%) | SPL (%) | False Positive Rate (%) ↓ |
> | :---: | :---: | :---: | :---: |
> | 0 (w/o EEC) | 63.5 | 31.2 | 15.2 |
> | 5 |  65.2 | 31.8 | 13.3 |
> | **10 (Default)** | **66.1** | **32.1** | **12.8** |
> | 20 | 66.3 | 32.1 | 12.7 |
>
> Introducing even a small memory capacity ($N_{max}=5$) yields a sharp initial drop in the False Positive Rate (FPR) from 15.2% to 13.3%. This explicitly proves the EEC's immediate effectiveness in intercepting erroneous premature stops.
> The FPR reaches its optimal point at 12.8% ($N_{max}=10$), after which performance strictly saturates (expanding to 20 yields a negligible 0.1% FPR gain).

---

> > ### Author Rebuttal · Reviewer_DNCd · 2026-04-03
> >
> > Dear authors,
> >
> > Thank you for the rebuttal. Thanks for the provided web link and the additional experiments. They have helped partially resolved my consern.
> >
> > I still have reservations for the use of LLM as a "logical filter". Using LLM and its "common sense reasoning" is a natural idea but it is not grounded in the knowledge of the scene. It overlooks the perception results and directly filter based on its logic. Does it make more sense to give LLM access to the images where objects are detected, and let the LLM decide based on not just the language prior but also the actual perception?
> >
> > Near equation 3, the elaboration reads: *"where fLLM(s) = 0 if the configuration exhibits internal logical conflicts (e.g., toilet appearing within living room)."* My question stems from here.

---

> > > ### Author Response · Authors · 2026-04-03
> > >
> > > We sincerely thank the reviewer for this highly insightful follow-up question.
> > >
> > > In PSG-Nav, the handling of the "bed in the living room" strictly depends on whether the bed is the Target Goal or merely a Contextual Object.
> > >
> > > 1. When the "bed" is the Target Goal: Visual Evidence overrides LLM Prior: If the user's instruction is literally "find the bed" and there is actually a bed in the living room, our system strictly prioritizes the visual observation over the LLM prior. when the target category is detected, the system bypasses the LLM completely. It relies exclusively on the continuous visual embedding of the crop to query the memory bank. If the visual features match past real-world experiences, it triggers the STOP action. The LLM (Eq. 3) has no authority to filter or veto this confirmation.
> > >
> > > 2. When the "bed" is a Contextual Object: If the robot is looking for a "television," but the perception model detects a "bed" in the living room, Eq. 3 will likely prune this bed hypothesis to maintain the topological coherence of the 3D-PSG. This is a deliberate design choice. In real-world zero-shot navigation, a "bed in the living room" detection is overwhelmingly more likely to be a perception error (e.g., the open-vocabulary detector misclassifying a sofa due to domain shift) rather than a rare anomaly. If we blindly trust raw contextual perception, the map becomes semantically corrupted, leading to widespread routing failures. Eq. 3 acts as a structural regularizer to absorb this systemic perception noise.
> > >
> > > 3. What if the Contextual Anomaly is Real? We fully agree that a robust system must respect rare but true physical observations. In reality, a true anomalous "bed in the living room" is rarely an isolated object; it is almost always accompanied by related visual cues (e.g., pillows, blankets, a nightstand). Our perception module groups these spatially proximal items into a single Group Node. The presence of these highly correlated visual items dynamically shifts the local room semantic distribution of that specific area (e.g., from living room to bedroom). When the LLM evaluates this configuration via Eq. 3, it receives a grounded, multi-object context (e.g., [bed, pillow] in a functional sub-area), rather than an isolated, illogical label. Because the grouped visual evidence internally supports the anomaly, the LLM recognizes the semantic coherence and retains the configuration. Thus, Eq. 3 acts as a filter for ungrounded perceptual noise, not a filter for grounded reality.
> > >
> > > 4. Why not give the LLM access to images? The open-vocabulary perception model has already grounded the image into semantic probability distributions (Eq. 1). Re-evaluating candidate object crops with a VLM at every step would severely bottleneck the system, destroying our 1.16Hz real-time efficiency.
> > >
> > > We deeply appreciate you pushing us on this critical aspect. It highlights the core philosophy of PSG-Nav: Vision strictly governs goal confirmation, while LLMs govern contextual routing. We will explicitly incorporate this "Target vs. Context" distinction, as well as the explanation of how Group Nodes preserve grounded anomalies, into the discussion of Eq. 3 to eliminate any ambiguity for future readers.

---

### Decision · Program_Chairs · 2026-04-30

**Decision:**

Accept (regular)

**Comment:**

This paper proposes to keep the distributional uncertainty from perception in constructing scene graphs instead of thresholding to the most likely classes, and then sampling from this distribution to generate plausible worlds, for uncertainty-aware decision-making.

Overall, the key concerns raised by the reviewers were the lack of details, some questions about the empirical results, and questions about novelty. The author response, as well as the reviewer follow-ups, indicate that the lack of details and much of the questions about the empirical results have been addressed. The remaining question is whether the contribution is significant to be a standalone conference paper. From the results and the system design, I think it looks like a reasonable systems contribution: even though the pieces are standard, the paper combines them together effectively, to solve a problem in a sound way, and shows compelling empirical results.

The authors should revise the final paper to incorporate the details that were requested by the reviewers and shared in the response.